

# Quantitative impacts of meteorology and precursor emission changes on the long-term trend of ambient ozone over the Pearl River Delta, China and implications for ozone control strategy

Leifeng Yang[1*], Huihong Luo[1*], Zibing Yuan[1], Junyu Zheng[2], Zhijiong Huang[2], Cheng Li[2], Xiaohua Lin[1], Peter K.K. Louie[3], Duohong Chen[4]

[1]School of Environment and Energy, South China University of Technology, Guangzhou 510006, China
[2]Institute for Environmental and Climate Research, Jinan University, Guangzhou 511443, China
[3]Hong Kong Environmental Protection Department, 5 Gloucester Road, Wan Chai, Hong Kong, China
[4]Guangdong Environmental Monitoring Center, 28 Modiesha Street, Guangzhou 510308, China

Correspondence: Zibing Yuan (zibing@scut.edu.cn) and Junyu Zheng (zhengjunyu_work@hotmail.com)

**Abstract**

China is experiencing increasingly serious ambient ozone pollution, including the economically developed Pearl River Delta (PRD) region. However, the underlying reasons for ozone increase remain largely unclear, leading to perplexity in formulating effective ozone control strategies. In this study, by developing a statistical analysis framework combining meteorological adjustment and source
apportionment, we examine quantitatively the impacts of meteorology and precursor emissions from within and outside the PRD on the evolution of ozone during the past decade. We found that meteorological condition has mitigated ozone increase, and its variation can account for at most 15% of annual ozone concentration in the PRD. Precursor emission from outside the PRD ('non-local') makes the largest contribution to ambient ozone in the PRD and shows a consistently increasing trend, while
that from within the PRD ('local') shows a significant spatial heterogeneity and plays a more important role during ozone episodes over southwestern. Under general conditions, the impact on northeastern is positive but decreasing, and on southwestern is negative but increasing. During ozone episodes, the impact on northeastern is negative and decreasing, while on southwestern is positive but decreasing. Central and western PRD is the only area with increasing local ozone contribution. The spatial
heterogeneity in both local ozone contribution and its trend under general conditions and ozone episodes are well interpreted by a conceptual model collectively taking into account ozone precursor emissions and their changing trends, ozone formation regimes, and the monsoonal and micro-scale synoptic conditions over different sub-regions of the PRD. In particular, we conclude that the inappropriate $NO_x$/VOC control ratio within the PRD over the past years is most likely responsible for the ozone
increase over southwestern, both under general conditions and during ozone episodes. By investigating the ozone evolution influenced by emission changes within and outside PRD during the past decade, this study highlights the importance of establishing a dichotomous ozone control strategy to tackle with general conditions and pollution events separately. $NO_x$ emission control should be further strengthened to alleviate peak ozone level during episodes. Detailed investigation is needed to retrieve appropriate
$NO_x$/VOC ratios for different emission and meteorological conditions, so as to maximize the ozone reduction efficiency in the PRD.

**Keywords:** Ozone, Meteorological adjustment, Empirical orthogonal function, Ozone formation regime, Pearl River Delta



## 1. Introduction

Thanks to a series of stringent air pollution control measures, most types of air pollutants, including $SO_2$, $NO_x$, CO, $PM_{10}$ and $PM_{2.5}$, exhibited decreasing concentrations in the past six years (2013-2018) in China, with the only exception of ozone (Souri et al., 2017; Lin et al., 2018; Lu et al., 2018; Koukouli et al., 2018;Wang et al., 2018; Zhang et al., 2018). During 2015-2018, ozone concentrations in the three major city clusters, Beijing-Tianjin-Hebei, Yangtze River Delta, and Pearl River Delta (PRD), had increased by 20%, 4%, and 14%, respectively (Report on the State of the Environment in China, http://english.mee.gov.cn/Resources/Reports/soe/). Although with comparable median ozone concentrations, the magnitude and frequency of high-ozone events are much higher in China than those in Japan, South Korea, Europe, and the United States (Lu et al., 2018). Ozone would become one of the major air pollution control targets in China in the near future to protect public health.

Ozone control is far more difficult than particulate matter (PM) control, according to the experiences in Los Angeles and Mexico City (Madronich, 2014). The difficulties of ozone control lie in two major aspects. First, ozone can be contributed by both local formation and non-local transport, and their relative importance is largely driven by meteorological conditions and precursor emission characteristics (Elminir, 2005; Beaver and Palazoglu, 2009; Kovač-Andrić et al., 2009). Moreover, ozone is a secondary pollutant with non-linear relationship with its precursors, $NO_x$ and volatile organic compounds (VOCs) (Stevenson et al., 2013; Thompson et al., 2014). Synergistic control with desirable VOC-to-$NO_x$ reduction ratio is required for ozone reduction. However, such a ratio is hard to determine and practically implement due to our limited understanding on VOC emissions, especially those fugitive (Ou et al., 2016). The appropriate VOCs-to-$NO_x$ reduction ratio may also vary greatly under different meteorological conditions. Therefore, from an ozone control point of view, it is fundamental to quantitatively understand the roles of meteorology and precursor emissions in shaping local and non-local ozone contributions, and their evolution during a long time scale in response to meteorology and emission changes.

Meteorology could either strengthen or dampen the efforts of precursor emission control on ozone reduction (Elminir, 2005; Beaver and Palazoglu, 2009; Kovač-Andrić et al., 2009). Hence, in order to investigate the effectiveness of precursor control during a long period, it is a common practice to homogenize meteorological conditions. In numerical simulation studies, such a philosophy is implemented by a set of scenarios in different meteorological and emission conditions (Gilliland et al., 2008; Godowitch et al., 2008; Wu et al., 2008; Foley et al., 2015). Differences in ozone levels between scenarios with the same meteorological conditions (emissions) are attributed to emission (meteorology) changes. Statistical models are also widely applied to establish relationship between ozone and meteorological variables so as to remove the meteorological impact, which is usually called meteorological adjustment (Lu and Chang, 2005; Zheng et al., 2007; Foley et al., 2015). After meteorological adjustment, ozone changes are solely attributed to emission changes.

Both local and non-local emission changes contribute to ambient ozone levels in a particular region. From an ozone control point of view, it is also essential to quantitatively differentiate local and non-local contributions. Source apportionment module coupled in chemical transport models, e.g. the Ozone Source Apportionment Technology (OSAT) in the Comprehensive Air-quality Model with extensions (CAMx) and the Integrated Source Apportionment Method (ISAM) in the Community Multiscale Air Quality (CMAQ), are widely used to attribute ambient ozone concentrations at a particular place into



different (local and non-local) source regions and categories (Li et al., 2013; Kwok et al., 2015). As
numerical simulation is suffered from uncertainties in emission inventories and largely constrained in
time span due to computing resources, statistical models, e.g. lowest-as-background method (Nielsen-
Gammon et al., 2005; Xue et al., 2014) and Empirical Orthogonal Function (EOF) (Langford et al., 2009;
Berlin et al., 2013), are preferentially adopted when the long-term monitoring data is available. They
apportion local and non-local contributions by examining variability and co-variability of ozone
concentrations at multiple monitoring sites. However, without meteorological adjustment, source
apportionment by both methods reflects only the absolute contribution from local and non-local sources
/ processes and cannot directly link with local and non-local emissions of ozone precursors. Therefore,
combined application of meteorological adjustment and source apportionment are indispensable in
investigating the effect of local and non-local emission changes on long-term ambient ozone variations.
Such combined application has not been reported in previous studies.

In this study, PRD is used as a research target area. After restraining its annual $PM_{2.5}$ concentration below
$35\mu g\ m^{-3}$ (China's National Ambient Air Quality Standard for annual $PM_{2.5}$) for four consecutive years
(2015-2018), PRD is the first major city cluster in China to transfer its main air pollution control target
onto ozone. By utilizing continuous ozone monitoring at multiple stations across the PRD since 2007,
we investigate the impacts of meteorology and local ('within PRD') and non-local ('outside PRD')
emission changes on the long-term trend of ambient ozone by using the framework of meteorological
adjustment followed by local and non-local contribution differentiation. Ozone contributions from
meteorology and local and non-local emissions are quantitatively demonstrated in 2016 and 2017, the
recent two years with significant ozone increase. We further develop a conceptual model depicting the
impact of emission control within the PRD to the ambient ozone, both under general conditions and
during ozone pollution episodes. Evaluation on the effectiveness of ozone precursor control measures
within and outside the PRD during the past decade would shed light on future control efforts that
hopefully shorten the ozone abatement paths experienced in Europe and the United States.

## 2.  Data and Method
### 2.1  Ozone and meteorological data set
Hourly ozone monitoring data at fifteen monitoring stations across the PRD from 2007 to 2017 are used
to calculate maximum daily 8-hour average (MDA8) in this study. Missing data are filled taking the
yearly, monthly, weekly and hourly mean into account, otherwise it is replaced by the ozone data at the
nearest monitoring station (Zheng et al., 2007). Geographical distribution of the monitoring stations is
illustrated in Fig. 1, and the latitudes / longitudes and the types of functional areas where the stations are
located are provided in Table 1.

The meteorological data during the same period, including daily maximum 2m temperature (T, ℃), daily
minimum relative humidity (RH, %), total net surface solar radiation (SSR, $J/M^2$), and 10m mean wind
direction and speed (u and v, m/s), are retrieved from the European Center for Medium-range Weather
Forecast (ECMWF) simulations for meteorological adjustment. Temporal and spatial resolution is 3-hour
and 0.125°×0.125°, respectively. Meteorological condition at the ozone monitoring station is represented
by the simulation data at the closest point to the station, as illustrated in Fig. 1. In this study, we composed
an ozone and meteorological dataset with 4018 days at fifteen stations.



### 2.2 Meteorological adjustment

In this study, a statistical analysis framework combining meteorological adjustment and source apportionment is developed to identify ozone changes attributable to meteorology and local and non-local emissions. Long-term trends of ozone changes by meteorological conditions and local and non-local emissions are subsequently evaluated by trend analysis. In this study, 'local' emissions refer to those from within the PRD, while 'non-local' emissions refer to those from outside the PRD. A conceptual diagram highlighting major calculation procedures of the statistical analysis framework is shown in Fig. 2.

In meteorological adjustment, Kolmogorov-Zurbenko (KZ) filter is firstly used to separate the raw ozone and meteorological data into long-term, seasonal and short-term data (Rao and Zurbenko, 1994a; Rao and Zurbenko, 1994b). KZ filter can be expressed as

$$X(t) = LT(t) + SE(t) + ST(t) \qquad (1)$$

Where $X(t)$ is the raw time series data, $LT(t)$ reflects the long-term trend in the time scale of years, $SE(t)$ is the seasonal variation in the time scale of months, and $ST(t)$ refers to short-term component in the time scale of days.

The KZ filter repeats the iterations of a moving average to remove the high-pass signal defined by

$$Y_i = \frac{1}{m} \sum_{j=-k}^{k} A_{i+j} \qquad (2)$$

where $k$ is the number of values included on each side, the window length $m=2k+1$, $i$ is interval time, $j$ is window variables, and $Y$ is the input time-series. Thus the output of the $i^{th}$ pass becomes the input for the $i+1^{th}$ pass, and so on. Different scales of motion are obtained by changing the window length and the number of the iterations (Milanchus et al., 1998; Eskridge et al., 1997). The filter periods of less than $N$ days can be calculated with window length $m$ and the number of iterations $p$, as

$$m \times p^{1/2} \leq N \qquad (3)$$

So a $KZ(15, 5)$ filter with the window length of 15 with 5 iterations remove cycles of 33 days. The ozone and meteorological time series by $KZ(15, 5)$ refer to their baseline variations which are the sum of long-term $LT(t)$ and seasonal components $SE(t)$.

$$BL(t) = KZ_{(15,5)} = LT(t) + SE(t) = KZ_{(36\ 53)} + SE(t) \qquad (4)$$

The long-term trend is separated from the raw data by $KZ(365, 3)$ with the periods >632d, and then the seasonal and the short-term component $ST(t)$ can be derived by

$$SE(t) = KZ_{(15,5)} - KZ_{(36\ 53)} \qquad (5)$$
$$ST(t) = X(t) - BL(t) = X(t) - KZ_{(15,5)} \qquad (6)$$

After KZ filtering, meteorological adjustment is conducted by stepwise regression between ozone concentration and meteorological factors such as T, RH and SSR (Flaum et al., 1996; Wise and Comrie, 2005; Papanastasiou et al., 2012).

$$A_{BL}(t) = a_{BL} + \sum b_{BLi} \cdot M_{BLi} + \epsilon_{BL}(t) \qquad (7)$$
$$A_{ST}(t) = a_{ST} + \sum b_{STi} \cdot M_{STi} + \epsilon_{ST}(t) \qquad (8)$$
$$\epsilon(t) = \epsilon_{BL}(t) + \epsilon_{ST}(t) \qquad (9)$$
$$A_{ad}(t) = \epsilon(t) + \sum b_{BLi} \cdot \overline{M}_{BLi} + \sum b_{STi} \cdot \overline{M}_{STi} + a_{BL} + a_{ST} \qquad (10)$$

Formula 7 and 8 are the multivariate regression models between baseline and short-term ozone and meteorological factors, respectively. $A_{ST}(t)$ and $A_{ST}(t)$ are the baseline and short-term components of ozone and $M_{BL}$ and $M_{ST}$ are the baseline and short-term components of meteorological factors. The parameters $a$ and $b$ are the fitted parameters and $i$ is time points (days). $\epsilon(t)$ is the residual term. The



average meteorological condition $\bar{M}$ of the same calendar date throughout 11 years is used as the base condition for that date, and the meteorological adjustment is conducted against the base condition. By doing so, the inter-annual variation of meteorology is removed while the annual variation is largely reserved. With the homogenized annual variation of meteorological conditions, $A_{ad}$ *(t)* in formula 10

represents the meteorologically adjusted ozone variations, and the difference between $X(t)$ and $A_{ad}$ *(t)* reflects the meteorological impact. It is noted that, by using the average meteorological condition as the base condition, the average ozone concentration during the 11 years keeps unchanged.

### 2.3  Source apportionment of ozone contributions from local and non-local emissions

In this study, EOF and absolute principal component scores (APCS) are applied to apportion meteorologically adjusted ozone concentration into local and non-local emission sources. EOF transforms a large number of variables into a new set of uncorrelated, orthogonal principal components (PCs). The few new variables contains the most information of the original variables, and the new variables represent different processes contributing to ambient ozone levels. Here we present a brief

description of EOF and APCS. Detailed information regarding the method can be found in Langford et al. (2009) and Berlin et al. (2013).

EOF analysis is performed on the correlation matrix from the meteorologically adjusted ozone data set (4018 days × 15 stations), without further rotation of the PCs. The first step is to normalize the ozone

data (Thurston and Spengler, 1985; Guo et al., 2004).

$$Z_{ik} = (C_{ik} - C_i)/S_i \qquad (11)$$

where $C_{ik}$ is the concentration of ozone in sample $k$ of the station $i$, $C_i$ is the arithmetic mean value of ozone in station $i$ and $S_i$ is the standard deviation.

$$Z_{ik} = L_{ip} \bullet P_{pk} \qquad (12)$$

$L_{ip}$ is loadings of EOF without rotation and $P_{pk}$ is scores.

Since the factor scores are normalized with the mean to be zero, true zero is calculated through introducing an artificial sample with the zero concentration. Then the APCS are estimated by subtracting the artificial sample from the true samples.

$$(Z_0)_i = \frac{(0-C_i)}{s_i} = -C_i/s_i \qquad (13)$$

$$(APCS)_{pi} = P_{pi} - P_{0pi} \qquad (14)$$

The regression between APCS and ozone concentration estimates source contributions to $C_i$ by

$$C_i = (b_0)_i + \sum APCS_p \divideontimes b_{pi} \qquad (15)$$

where $(b_0)_i$ is the constant term at station $i$, $b_{pi}$ is the coefficient of the source p, and $\sum APCS_p$ is the scaled value of the factor $p$. Multiplication of $\sum APCS_p$ and $b_{pi}$ calculates the contribution from source

p to ozone concentration. Local and non-local sources are determined according to the temporal and spatial distribution characteristics of source contributions across the PRD.

### 3.  Results and Discussion

### 3.1  Long-term trend of meteorological impact on ozone concentration

Fig. 3a shows the long-term trends of ambient ozone, meteorologically adjusted ozone, and the meteorological impact in the PRD during 2007-2017. Ambient ozone concentration in the PRD increased from 76 μg m$^{-3}$ in 2007 to 89 μg m$^{-3}$ in 2017, corresponding to an annual increase rate of 1.2 μg m$^{-3}$. Previous studies also evidenced ozone increase in the PRD (e.g. Li et al., 2014) and we here demonstrate



that such an increase has been continuing for more than a decade. After meteorological adjustment, ozone
concentration increases from 68 μg m$^{-3}$ in 2007 to 90 μg m$^{-3}$ in 2017, corresponding to an annual increase
rate of 2.0 μg m$^{-3}$. Higher increase rate of meteorologically adjusted ozone implies that if the
meteorological condition keeps unchanged throughout the 11 years, ambient ozone concentration would
increase more significantly. As shown in Fig. 3a, meteorological conditions are generally favorable for
ozone pollution during 2007-2011, responsible for at most 6 μg m$^{-3}$ of ozone increase. During 2012-2017,
meteorological condition became unfavorable for ozone pollution, leading to at most 6 μg m$^{-3}$ of ozone
reduction. Comparing between the most favorable (2007) and unfavorable (2016) year, meteorological
condition change ozone concentration by 12 μg m$^{-3}$ at most, roughly corresponding to 15% of annual
ozone concentration.

It should be noted that meteorological adjustment does not change the overall increasing trend of ozone
concentration, indicating that emission change is the primary driving factor for the long-term ozone trend.
However, as shown in Fig. 3a, the fluctuation of ozone concentration is suppressed by meteorological
adjustment, indicating that meteorology plays an important role in the inter-annual fluctuation of ozone
concentration, especially during 2011-2015 when ozone changes due to emissions are minor. During
some specific period, meteorology plays a greater role in governing ozone changes than emissions, such
as during 2016-2017 as will be discussed in section 3.4. As shown in Fig. 3b, meteorological adjustment
significantly reduces the magnitude of ozone spikes, indicating that meteorological condition is the most
important driving factor for ozone episodes in the PRD.

Fig. 3c shows the impacts from different meteorological factors (T, RH, SSR, u and v) on ozone
concentration. Overall, SSR is the most crucial factor and their variation follows well with that of the
total meteorological impact. Contribution from the other four factors are comparable and relatively
insignificant. As expected, higher SSR, higher T and lower RH are favorable for ozone production.

**3.2 Spatial distribution of meteorological impact**
We further examine the spatial distribution of meteorological impact. Fig. 4a shows the spatial
distribution of averaged ozone concentration in the PRD and the annual ozone concentration changes
before and after meteorological adjustment. Although northeastern PRD has the overall highest ozone
concentrations, central and western PRD shows the most rapid ozone increase during the 11 years (black
bars), and such increases are further substantiated if meteorological impact is removed (green bars).
There are two sub-regions in the PRD with overall decreased ozone concentrations, one in the northeast
(TH and JGW) and the other in the southwest (ZML and TJ). The ozone decrease is largely mitigated or
at ZML even reversed after meteorological adjustment. The different mechanisms leading to the ozone
increase in these two sub-regions are explained by a conceptual model in section 3.6.

The spatial distribution of meteorological impact in each year during 2007-2017 is illustrated in Fig. 4b.
It is noted that when the meteorological condition favors ozone pollution in the PRD, it increases more
in the central and western area. On the contrary, when it decrease ozone concentration in the PRD, central
and western PRD is also the region with larger decrease in most years. Therefore, central and western
PRD is a meteorology-sensitive region for ozone pollution. Formulation of ozone control strategy in this
region needs to consider meteorological impact.



### 3.3  Identification of ozone changes resulted from local and non-local emissions

Long-term variation of meteorologically adjusted ozone reflects the impacts from precursor (VOCs and
$NO_x$) emission changes. As ozone can be contributed by both local production and long-range transport,
it is important to quantitatively separate them from an emission control point of view. Considering PRD's
monsoonal synoptic condition and that most local emissions are concentrated in the central PRD area,
local ('within PRD') emissions tend to pose distinct impacts to different sub-regions in different seasons
while the impacts from non-local ('outside PRD') emissions in a relatively larger scale tend to be spatially
consistent over the PRD. We use this philosophy to examine the PCs derived from EOF analysis.

According to the Kaiser's rule (Wilks, 2006), three PCs are retained in EOF analysis, explaining 53%,
16% and 7% of total variance, respectively. Fig. 5a shows the interpolated PC loadings in the PRD, and
Fig. 5b shows the long-term variation of PC scores during the 11 years. PC1 shows relatively consistent
spatial distribution across the PRD, with its loadings ranging from 0.47 at TH to 0.84 at JJZ. Further
examination on the relationship between PC scores and wind direction discovered that the score of PC1
is higher during high ozone concentration in the PRD, and is associated with northeasterly wind (Fig.
6a). Situated along the southeastern coast of China, PRD has two main prevailing winds, northeasterly
mainly during winter and spring and southwesterly mainly during summer and fall. Northeasterly wind
tends to bring emissions from inland to the PRD, while southwesterly wind originated from the ocean is
relatively clean. All the above evidences support the notion that PC1 is associated with non-local impact
from continental long-range transport. Higher impact in the central PRD may be caused by the rough
land use and micro-scale circulation in this urbanized region that increases the residence time of non-
local ozone. The score of PC1 (Fig. 5b) is almost consistently increasing during the 11 years, indicating
increased ozone contribution from long-range transport.

In comparison, PC2 and PC3 loadings show significant spatial variations. PC2 loadings have an obvious
north-south gradient with different signs, indicating that the impact of PC2 on northern and southern
PRD are reversed at all times. Further examination on their relationship with wind direction, as shown
in Fig. 6b and 6c, indicates that during high ozone periods, PC2 score tends to be positive with southerly
winds and negative with northerly winds. With southerly winds, northern PRD receives the highest
impact from PC2, leading to increased ozone concentration. On the contrary, southern PRD receives the
highest impact (negative score and negative loading) with northerly winds. This reflects exactly the
impact from emissions within the PRD posed by the north-south components of the prevailing winds.
Similarly, PC3 is associated with the impact from local emissions by the west-east components of the
prevailing winds. Therefore, PC2 and PC3 collectively reflect the impact of local emissions on ozone
formation. PC2 and PC3 scores show a bimodal pattern that are higher in 2007 and 2011-2014 (Fig. 5b).
This suggests that local emissions pose higher ozone contribution to northern and eastern PRD during
2007 and 2011-2014 while to southern and western PRD during 2008-2010 and 2015-2017.

The PC loadings and scores may reflect the spatial distributions and temporal variations of the PCs,
respectively. However, as they are normalized values, APCS calculation is conducted to quantify the
absolute ozone contributions from local and non-local emission sources. Fig. 7 shows ozone
contributions from local and non-local emission sources at three representative stations, TH in northern,
LH in central and DH in southern PRD. A first glance on this figure reveals identical and consistently
increasing non-local trends at all three stations. Non-local emission contributions at DH reach 90~115



µg m⁻³, more than doubling those of 44~56 µg m⁻³ at TH. As explained previously, such a spatial heterogeneity is mainly caused by longer residence time of non-local ozone in the urbanized area. In comparison, local emission contributions show differences in both magnitude and trend over three stations. Local emission contribution on ozone ranges from 15~30 µg m⁻³ at TH, 1~6 µg m⁻³ at LH, and -30~-15 µg m⁻³ at DH. As a net effect of ozone production and loss, the positive or negative sign of local emission contribution reflects the relative strengths of ozone production by $HO_x$ and $RO_x$ cycles and ozone loss by NO titration and deposition. As non-local emission contributions dominate over the local counterpart at all stations, its consistently increasing trend determines the meteorologically adjusted ozone trend over the 11 years.

We further plot the spatial distribution of ozone contribution from local and non-local emissions and its long-term changes over the PRD, as shown in Fig. 8. Local emissions give positive contribution to northeastern, with the largest contribution of 31 µg m⁻³ at JGW, and negative contribution to southwestern, with the largest contribution of -23 µg m⁻³ at DH. Furthermore, apart from reversed ozone contribution from local emissions, northeastern and southwestern PRD also exhibit reversed trends in changes of ozone contribution from local emissions during the 11 years, as illustrated in the bars of Fig. 8. The most significant increase trend is found over southwestern, with the largest increase rate of 0.6 µg m⁻³ year⁻¹ at DH, while the most significant decrease trend is found over northeastern, with the largest decrease rate of 0.8 µg m⁻³ year⁻¹ at JGW. The underlying mechanism resulting in the opposite trends in both local ozone contribution and its long-term changes between northeastern and southwestern are explained with a conceptual model in section 3.6. In comparison, the ozone contributions from non-local emissions are relatively consistent over the PRD, and non-local emission poses increasing influence on ozone for the entire region.

### 3.4 Identification of driving factors for ozone changes in 2016 and 2017

With meteorological adjustment and source apportionment, the contributions from meteorology and local and non-local emissions to the ambient ozone changes can be quantitatively analyzed for all years at all stations. In this section, we demonstrate this capability by analyzing the relative importance of meteorology and local and non-local emissions to the ozone changes in the recent two years, 2016 and 2017. Significant ozone level increases are revealed at most stations during the two years, with the average concentration rising from 81 µg m⁻³ in 2016 to 87 µg m⁻³ in 2017. It is found that meteorology, local emission and non-local emission contribute to around 3.5 µg m⁻³, -0.1 µg m⁻³ and 2.0 µg m⁻³ of ozone increase, respectively. Overall, meteorology plays a greater role in elevating ozone levels during these two years.

Contributions from meteorology and local and non-local emissions are further analyzed at each monitoring station, as listed in Table 2. Under general conditions, in comparison with local and non-local emissions, meteorology gives the highest contributions to ozone changes at all stations except for CZ and DH, the two southwestern-most stations. In addition, local emissions gives higher contributions than non-local ones at CZ, DH, JJZ, ZML and TJ, the cluster of stations in the southwestern PRD. Therefore, the ozone increase over southwestern PRD during these two years is most attributable to local emission changes, while the ozone increase in other parts of the PRD is firstly driven by meteorological condition changes, followed by non-local emission changes. This suggests that in order to reduce ozone levels in the southwestern PRD, strengthening local VOCs emission control should be of the top priority, so as to





prevent ozone titration from decreasing further.

### 3.5  Impact of meteorology and emission changes during ozone episodes

In this section, we examine the impacts of meteorology and local and non-local emission changes to ambient ozone level during ozone episodes. Ozone episodes are defined as days with MDA8 ozone concentration greater than 160 µg m$^{-3}$ at five stations or more across the PRD. As shown in Fig. S1, ozone levels are the highest in the central PRD, mainly Guangzhou and Foshan, during ozone episodes.

Fig. S2 shows the long-term trends of ambient ozone, meteorologically adjusted ozone, and the meteorological impact in the PRD during ozone episodes in 2007-2017. Ambient ozone concentration during episodes increases from 150 µg m$^{-3}$ in 2007 to 161 µg m$^{-3}$ in 2017, corresponding to an annual increase rate of 1.0 µg m$^{-3}$. It is noteworthy that meteorological adjustment does not alter ozone concentration much, with the largest change of 3 µg m$^{-3}$ only. This implies that, although with significant variation under general conditions, meteorology does not vary significantly during ozone episodes across all years. Changes in precursor emissions are therefore the driving factor for long-term ozone variations during ozone episodes. A slightly different picture is discovered in 2017 during which meteorology is the major culprit for ozone increase. Without meteorological impact, ozone level during episodes should be lower than that in 2016.

We further differentiate ozone changes into those by local and non-local emissions using EOF/APCS approach. Four principal components are discovered, and they are assigned to local or non-local emissions by their spatial variations, as shown in Fig. S3. Fig. 9 illustrates the long-term trend of ozone contribution by local and non-local emissions during ozone episodes at TH, LH and DH stations. At TH and LH, non-local emissions give dominant contribution to ozone, while local emissions pose negative impacts, while contributions from local and non-local emissions are comparable at DH. Different from general conditions during which non-local contribution shows a consistently increasing trend, non-local contribution fluctuates greatly during ozone episodes and presents a bimodal picture. Starting from 2014, ozone contribution from non-local emissions has leveled off and decreased gradually.

Local emission contribution to ozone during episodes differs greatly in different areas. As shown in Fig. 10a, local emissions give positive contribution to southwestern, with the largest average contribution of 78 µg m$^{-3}$ at DH. They pose negative contribution to northeastern, with the largest contribution of -36 µg m$^{-3}$ at TH and HG. Such a spatial distribution is contrary to that during general conditions, as illustrated in Fig. 8a. Stations over central PRD show increasing trend, with the largest increase rate of 1.9 µg m$^{-3}$ year$^{-1}$ at HG, while stations surrounding central and western PRD show decreasing trend, while the largest decrease rate of 3.5 µg m$^{-3}$ year$^{-1}$ at JGW. In comparison, ozone contributions from non-local emissions are relatively consistent over the PRD, with the hotspot shifted from central and western PRD under general conditions (Fig. 8b) to central and eastern PRD during ozone episodes (Fig. 10b). The entire PRD experienced increasing ozone contribution from non-local emission. Comparing with non-local emission, ozone contribution from local emission and its trend show significant spatial heterogeneity. We develop a conceptual model to explain in detail the underlying mechanisms resulting in the distinct spatial distribution of local ozone contribution and its trend between general conditions and ozone episodes, as elaborated in section 3.6.



### 3.6 A conceptual model describing impact of local emission changes to ozone in the Pearl River Delta

As discussed in section 3.3 and 3.5, the spatial pattern of ozone contribution from local ('within PRD') emissions and its long-term changes in the PRD under general condition and during ozone episodes present different pictures. Under general condition, local emissions give positive and decreasing contribution to ozone over northeastern PRD, and negative and increasing contribution over southwestern (Fig. 8a). In contrast, during ozone episodes, local emissions give negative and decreasing contribution over northeastern, and positive and decreasing contribution over southwestern. Central and western PRD is the only region having slight increasing local ozone contribution during episodes (Fig. 10a). In this section, we aim to provide detailed explanation on such phenomena by developing a conceptual model collectively taking into account ozone precursor emissions and their changing trends, ozone formation regimes, and the monsoonal and micro-scale synoptic conditions over different sub-regions of the PRD.

### 3.6.1 General condition

PRD has distinct VOCs and $NO_x$ emission characteristics across its different sub-regions, leading to different prevailing ozone formation regimes (OFR) over the PRD. Central PRD, essentially western and southern Guangzhou, Foshan and western Dongguan, is the area with the most significant economic and industrial activities. Central PRD is associated with significant amount of VOCs and $NO_x$ emissions (Zheng et al., 2009, Zhong et al., 2018), and is mostly in a VOC-limited OFR (Ye et al., 2016). The polluted air mass can be transported to different areas of the PRD under different prevailing winds, and largely determines the ozone behaviors over those areas. In the past years, $NO_x$ emissions are decreasing due to stringent control measures, while VOCs emissions are increasing, as shown in Fig. S4.

Northeastern PRD is mainly a rural area with plenty of vegetation coverage. Significant VOC emissions from biogenic sources make it primarily in a $NO_x$-limited OFR, especially in summer (Ye et al., 2016). In summer and fall, southwesterly winds originated from the South China Sea prevail, bringing the $NO_x$-laden air mass from central PRD to the downwind $NO_x$-limited northeastern and increasing ozone levels over TH, XP and JGW stations. However, $NO_x$/VOC ratio in the air mass is decreasing during the past years due to emission control measures that are preferentially targeting on $NO_x$ emissions. Lowered $NO_x$/VOC ratio would inhibit ozone production in the $NO_x$-limited northeastern, leading to a downward ozone trend. In contrast, southwestern PRD shows relatively higher $NO_x$/VOC emission ratios, and is mostly in a VOC-limited OFR (Ye et al., 2016). The OFR would shift to be more VOC-limited in winter due to the suppressed biogenic VOC emissions and reduced reaction rate of $HO_x$ and $RO_x$ cycles. In winter and spring, northeasterly winds originated from the Eurasia Continent prevail, bringing the $NO_x$-laden air mass from the central PRD to the southwestern. The $NO_x$-laden air mass would react preferentially with ozone in the VOC-limited southwestern, thereby decreasing the ozone levels at CZ, DH and ZML stations. Due to the strengthened $NO_x$ emission control that reduces $NO_x$/VOC ratio from the central PRD, ozone titration is largely mitigated, leading to an upward ozone trend over southwestern in the past few years. Fig. 11 provides a conceptual diagram on the impact of local emission control on ozone concentrations and their changing trends over the PRD.

Hence, the combined influences by reduced ozone titration from local emissions and increased ozone import from non-local emissions make southwestern the area having the most rapid ozone increase over the PRD. In order to curb ozone increase in the southwestern, VOC emission control within the PRD



must be strengthened to elevate $NO_x$/VOC ratio into a level that ozone titration would not be further
reduced. With decreased influence from local emissions, northeastern shows the least ozone increase.

### 3.6.2 Ozone pollution episodes

Both meteorology and precursor emissions exhibit significant differences during ozone episodes in
comparison with general conditions. Ozone episodes typically happen in summer and fall with hot and
sunny weather and weakened background wind, which is very often associated with a high pressure ridge
or approaching of a tropical cyclone (Huang et al., 2006). Temperature very often rises above 30 degree
Celsius with abundant sunshine, leading to more intense biogenic VOC emissions over the PRD.
Considering $NO_x$ emissions are insensitive to temperature rise and the high reactivity of biogenic VOCs,
the effective $NO_x$/VOC ratio becomes much lower. As a result, $NO_x$-limited OFR over northeastern is
intensified, and VOC-limited OFR over southwestern shifts to $NO_x$-limited. VOC-limited area shrinks to
merely central PRD and the magnitude is largely weakened (Wang et al., 2011; Jin and Holloway, 2015).
Due to significant $NO_x$ emissions, the urban central PRD is probably the last area turning into $NO_x$-
limited due to enhanced biogenic VOC emissions during ozone episodes.

In addition, the prevailing wind direction changes from northeasterly / southwesterly to easterly, as shown
in Fig. S5. With weakened background wind, micro-scale circulations such as land-sea breeze develop
around the Pearl River Estuary (PRE), and becomes an effective mechanism in trapping and mixing up
pollutants emitted surrounding the PRE (Lo et al., 2006). Micro-scale circulations increase the residence
time of pollutants over the PRE and thus expedite chemical reactions to produce ozone. High ozone
produced around the PRE is brought to southwestern PRD (a 'sink' region) by the weak easterly wind,
thereby increasing ozone levels at DH, ZML and TJ stations. In contrast, with easterly wind, northeastern
receives little impact from the central PRD ozone hotspot while instead serves as a 'source' region (ozone
import from further east is accounted for as impact from non-local emissions), thereby providing negative
contribution at TH, XP, JGW, HG and LH stations.


With higher biogenic VOC emissions and VOC oxidation rate, the OFR distribution over the PRD during
ozone episodes vary from that under general conditions. The preferential $NO_x$ emission reduction due to
stringent control would lead to downward trend of local ozone contribution over northeastern due to
intensified $NO_x$-limited OFR, also downward trend over southwestern due to shift from VOC-limited to
$NO_x$-limited OFR. An upward trend is only discovered over central and western PRD (HG, LH, HJC and
ZH) where $NO_x$ emissions are very strong and still persist in VOC-limited OFR. Fig. 12 provides a
conceptual diagram on the impact of local emission control on ozone concentrations and their changing
trends over the PRD during ozone episodes.

Hence, even with different formation mechanisms from general conditions, southwestern PRD, mainly
Zhongshan, Zhuhai and eastern Jiangmen, is still the area with the most significant impact from local
emissions during ozone episodes. However, with less $NO_x$ emissions than central PRD, OFR over
southwestern has shifted from VOC-limited to $NO_x$-limited, leading to reduced local ozone contribution.
Comparison of different trends between central and southwestern PRD actually highlights the fact that
$NO_x$ emission control is one of the possible means to reduce ozone levels over the PRD, especially during
ozone episodes with significantly enhanced biogenic VOCs emissions. Further reduction of $NO_x$
emissions, after bypassing the optimal effective $NO_x$/VOC ratio, would rapidly pull down peak ozone



level and eventually bring it into attainment (Ou et al., 2016). Different OFR characteristics under general condition and during ozone episodes also highlight the importance of formulating dynamic control
measures tailored for different emission and meteorological conditions.

### 4. Conclusion and Implication

Ambient ozone level in a particular area is determined by the interaction between meteorology and emission of ozone precursors, VOCs and $NO_x$. Differentiation of their impacts are important to evaluate
the effectiveness of emission control measures in the past and to shed light on directions for future control plans. In this study, we develop a statistical analysis framework to identify ozone changes attributable to meteorology and local and non-local emissions in the PRD. The framework is essentially a combination of meteorological adjustment and source apportionment by EOF. We found that meteorology does not alter the increasing trend of ozone during 2007-2017, but significantly mitigate the magnitude of
increasing. Ozone increase solely due to precursor emission changes would have been more significant.

In comparison with non-local precursor emissions, the impacts of local precursor emissions on ambient ozone present significant spatial and temporal heterogeneity over the PRD. Northeastern and southwestern exhibit different net ozone production and loss characteristics under general conditions and
during ozone episodes. In response to the preferential $NO_x$ emission control during the past years, local ozone contribution decreases over northeastern and increases over southwestern under general conditions, while decreases over both northeastern and southwestern but increases over central and western PRD during ozone episodes. Such a complex characteristics can be well interpreted by a conceptual model collectively taking into account ozone precursor emissions and their changing trends, ozone formation
regimes, and the monsoonal and micro-scale synoptic conditions over different sub-regions of the PRD. In particular, OFR shift during ozone episodes in response to higher biogenic VOC emissions and VOC oxidation rate is the fundamental cause for different trends both spatially and temporally. We conclude that the past control measures preferentially targeted on $NO_x$ are most likely responsible for ozone increase in the PRD, especially over southwestern by reduced ozone titration. However, OFR has started
to shift from VOC-limited to $NO_x$-limited over southwestern, especially during ozone episodes. Therefore, $NO_x$ emission control should be further strengthened to alleviate peak ozone levels.

By investigating the ozone evolution influenced by emission changes within and outside PRD during the past decade, this study highlights the complexity in ozone pollution control in the PRD. The complexity
lies in three aspects. First, ozone control should be location-specific. Northeastern is the area benefited from current control measures in the PRD, and the main focus should be on co-prevention and co-control with further northeastern areas, e.g. Jiangxi and Fujian, to reduce long-range transport; Central and southwestern PRD should pay more efforts on VOCs control to elevate $NO_x/VOC$ ratio into a level that ozone titration would not be further reduced. Second, ozone control should be temporally dynamic and
largely dependent upon meteorological conditions. OFR may change greatly under different meteorological conditions which would influence effective control strategy and deserve more in-depth investigation. In particular, precursor emissions surrounding the PRE should be preferentially controlled during ozone episodes as they may contribute greatly to ozone formation when trapped over PRE by the micro-scale circulations. They are responsible for ozone hotspot over southwestern with a drastically
increasing trend. Last but not least, under every circumstance, the most desirable $NO_x/VOCs$ ratio for emission control should be investigated in detail. For example, control measures during ozone episodes



should preferentially target on $NO_x$ in the context of significantly enhanced biogenic VOCs emissions. Comparison of different trends between central and southwestern PRD provides a perfect highlight on the effect of $NO_x$ control. Further reduction of $NO_x$ emissions, after bypassing the optimal effective

$NO_x$/VOC ratio, would rapidly pull down peak ozone level and eventually bring it into attainment (Ou et al., 2016).

**Authorship Contribution Statement**

ZY and JZ designed the experiments and LY, HL and XL carried them out. PKKL and DC provided ozone

monitoring data and contributed to the discussion of the results. LY and ZY drafted the paper, with all co-authors contributing to subsequent enhancements.

*These authors contribute equally to this article.

**Acknowledgements**

This work is supported by National Natural Science Foundation of China (No. 91644221) and the National Key Research and Development Program of China (No. 2016YFC0202201). The authors are grateful to Guangdong Environmental Monitoring Center and Hong Kong Environmental Protection Department for providing ozone monitoring data over the PRD for use in this study.






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






**Table 1. Location of fifteen ozone monitoring stations across the Pearl River Delta and their environmental background.**

| Station | Full name | City | Longitude (E) | Latitude (N) | Environmental Background |
|---------|-----------|------|---------------|--------------|--------------------------|
| CW | Central/Western | Hong Kong | 114.15 | 22.28 | Residential/Commercial |
| CZ | Chengzhong | Zhaoqing | 112.47 | 23.05 | Residential/Commercial |
| DH | Donghu | Jiangmen | 113.08 | 22.59 | Urban |
| HG | Haogang | Dongguan | 113.73 | 23.03 | Residential/Commercial |
| HJC | Huijingcheng | Foshan | 113.10 | 23.00 | Residential/Commercial |
| JGW | Jinguowan | Huizhou | 114.38 | 22.93 | Residential |
| JJZ | Jinjuzui | Foshan | 113.26 | 22.81 | Suburban |
| LH | Luhu | Guangzhou | 113.28 | 23.15 | Urban |
| LY | Liyuan | Shenzhen | 114.09 | 22.55 | Urban |
| TC | Tung Chung | Hong Kong | 113.91 | 22.27 | Residential |
| TH | Tianhu | Guangzhou | 113.62 | 23.65 | Rural |
| TJ | Tangjia | Zhuhai | 113.58 | 22.34 | Commercial/Industrial |
| XP | Xiapu | Huizhou | 114.40 | 23.07 | Commercial |
| YL | Yuen Long | Hong Kong | 114.02 | 22.44 | Residential |
| ZML | Zimaling | Zhongshan | 113.40 | 22.50 | Residential/Commercial |

**Table 2. Contributions of meteorology and local and non-local emission changes to the ozone concentration change (μg m$^{-3}$) in 2016-2017 at fifteen monitoring stations in the Pearl River Delta under general conditions.**

| Station | CZ | DH | JJZ | ZML | TJ | HJC | LH | LY |
|---------|-----|-----|-----|-----|-----|-----|------|-----|
| Meteorology | 1.9 | 3.1 | 3.3 | 4.0 | 5.3 | 3.2 | 4.0 | 3.2 |
| Local | 3.5 | 4.8 | 2.9 | 3.4 | 2.2 | 2.2 | -0.3 | 0.1 |
| Non-local | 2.2 | 2.7 | 2.8 | 2.6 | 2.0 | 2.5 | 2.3 | 1.5 |
| **Station** | **YL** | **TC** | **CW** | **HG** | **XP** | **JGW** | **TH** | **PRD** |
| Meteorology | 3.8 | 1.9 | 2.9 | 3.9 | 4.2 | 4.5 | 4.0 | 3.5 |
| Local | 0.4 | 0.3 | -0.2 | -2.7 | -5.5 | -6.3 | -5.0 | -0.1 |
| Non-local | 1.8 | 0.9 | 1.4 | 2.4 | 1.8 | 1.6 | 1.3 | 2.0 |



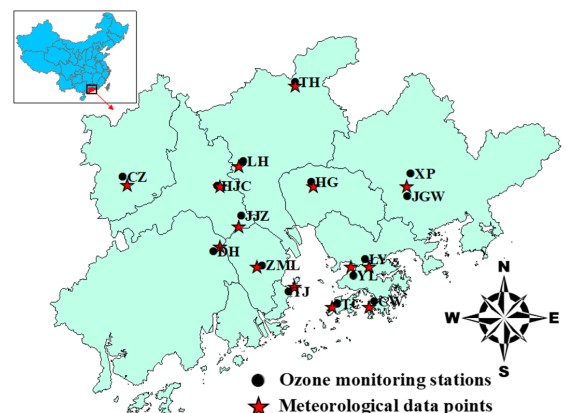


**Fig 1. Distribution of ozone monitoring stations and meteorological data points in the Pearl River Delta.**

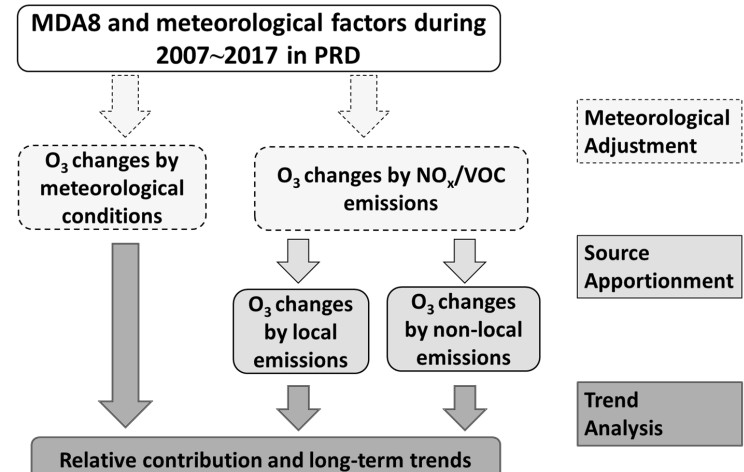

**Fig 2. Flowchart of the statistical analysis framework to identify the long-term impacts of meteorology and**

**local and non-local emissions on ambient ozone.**



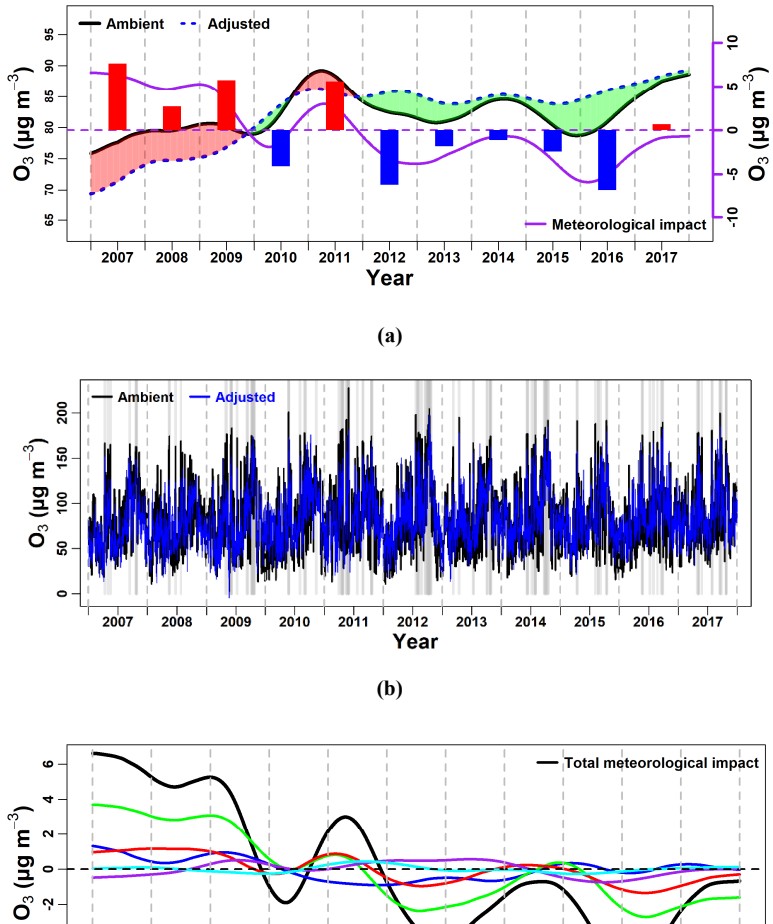

**Fig 3. (a)** Long-term trends of ambient ozone, meteorologically adjusted ozone, and the meteorological impact in the Pearl River Delta during 2007-2017. Periods with positive and negative meteorological impacts are shadowed in red and green, respectively. Red and blue bars represent ozone increase and reduction attributed to meteorology in each year, respectively. **(b)** Ozone concentration time series before (black) and after (blue) meteorological adjustment. Gray areas represent periods with ozone concentration over 160 μg m⁻³. **(c)** Long-term variations of meteorological impact by different meteorological factors.




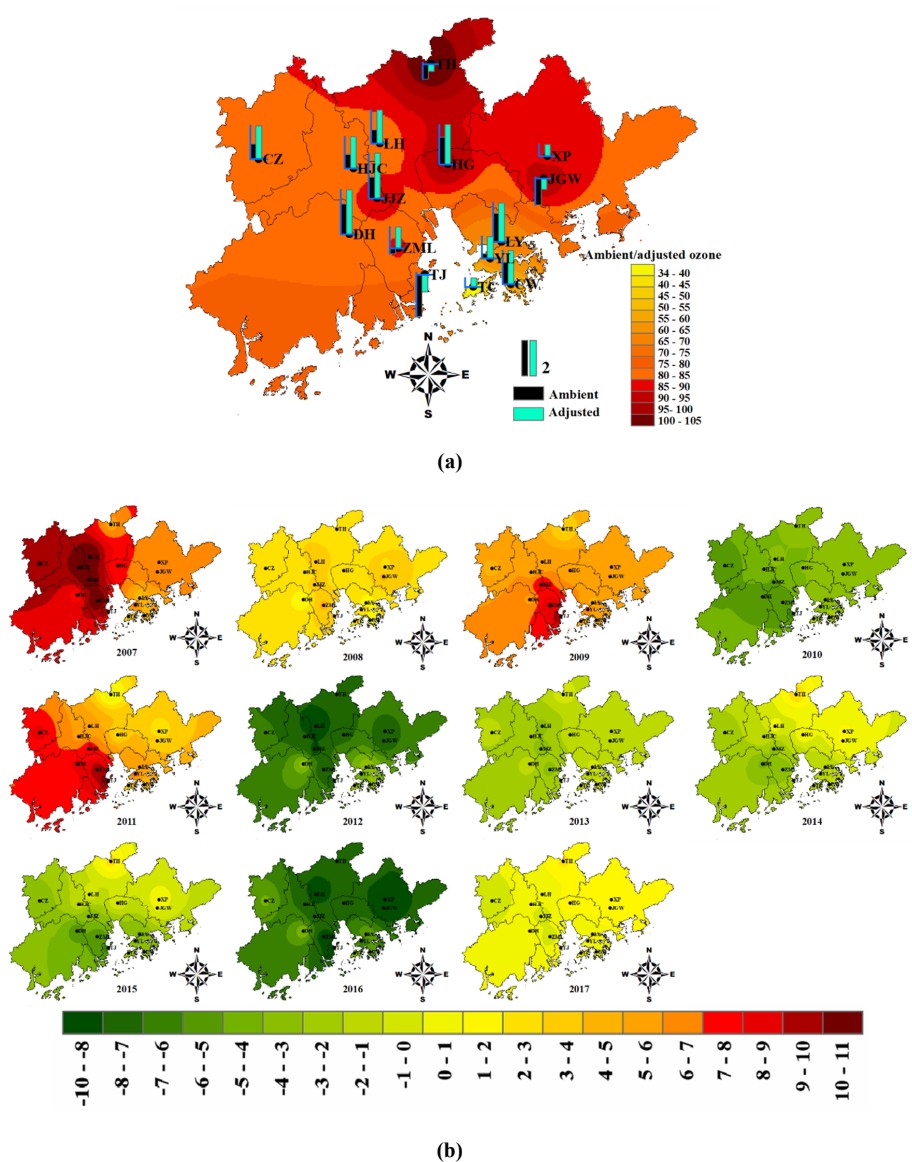

**Fig 4. (a)** Spatial distribution of averaged ozone concentrations (μg m$^{-3}$) in the Pearl River Delta and annual ozone changes (μg m$^{-3}$ year$^{-1}$) before and after meteorological adjustment over the fifteen monitoring stations during 2007-2017. The bar length in the legend corresponds to an annual increase of 2 μg m$^{-3}$. **(b)** Annual variation of meteorological impact on ozone concentration (μg m$^{-3}$) over the Pearl River Delta during 2007-2017.





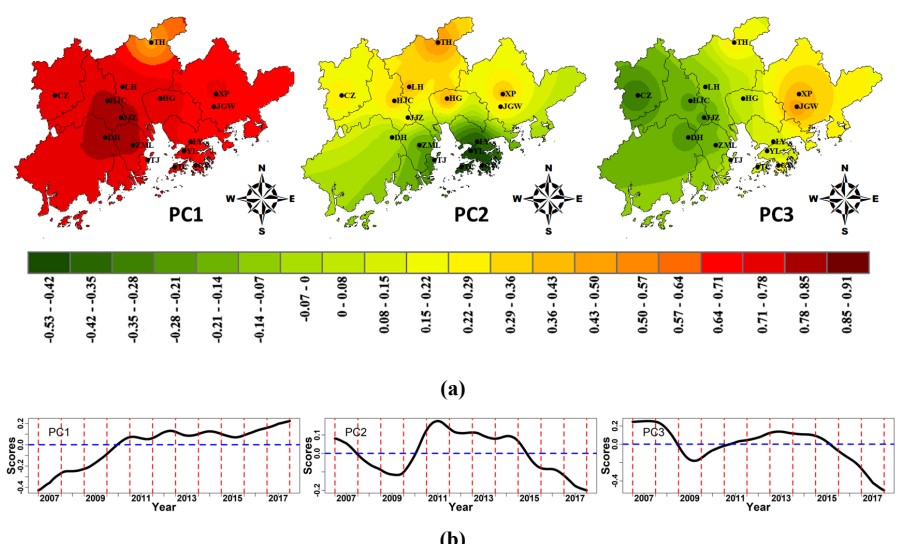

**730**

**Fig 5. (a) Spatial distribution of principal component loadings in the Pearl River Delta, and (b) long-term variation of principal component scores during 2007-2017. PC1 reflects non-local emission impacts while PC2 and PC3 refer to impacts from different local emissions.**

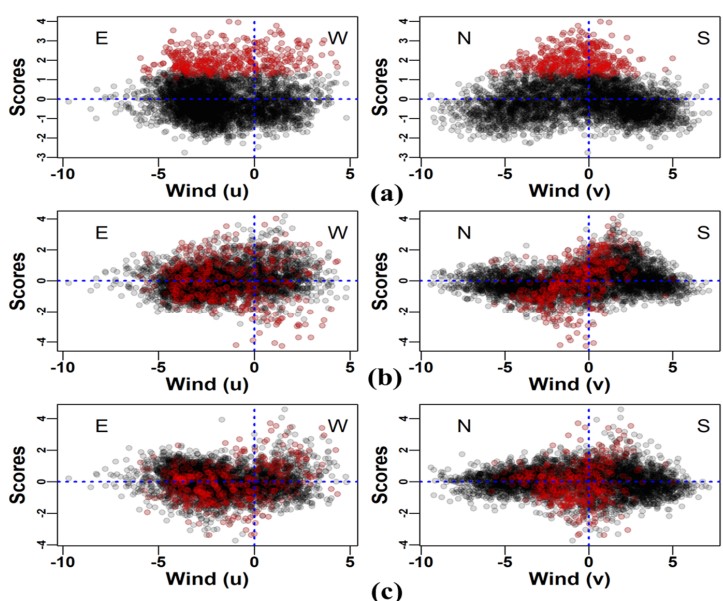

**735**

**Fig 6. Scatterplot between principal component scores (a-c: PC1-3) and wind (u and v). u and v are the east-west and north-south components of wind (u: +west/-east, v: +south/-north). Red points refer to samples with high ozone concentration (over 90[th] percentile).**




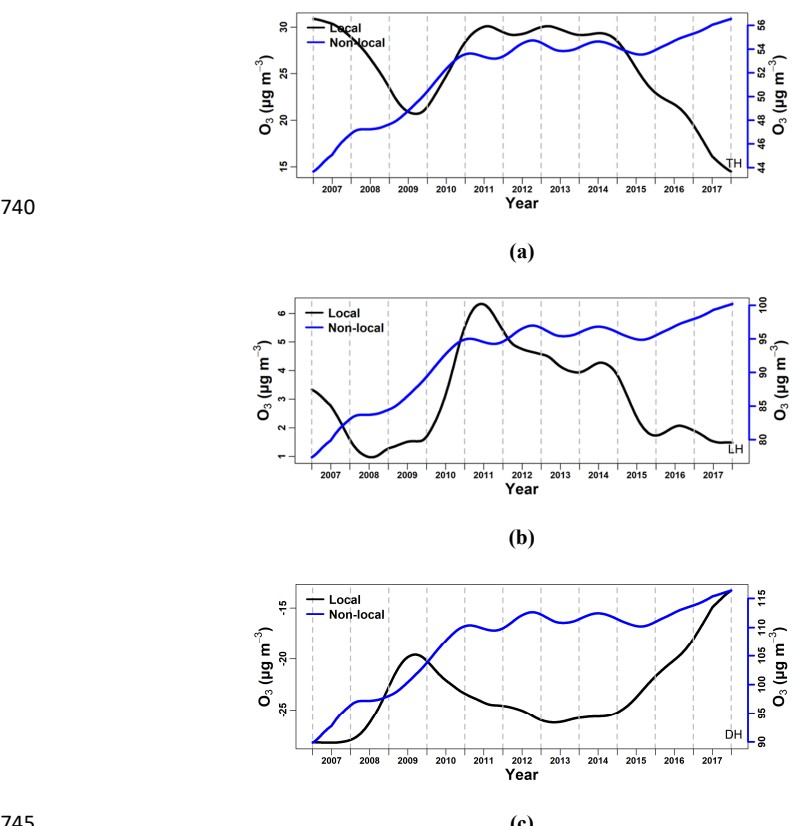

**Fig 7. Long-term trend of ozone contributed by local (black) and non-local (blue) emission sources from 2007 to 2017 at (a) TH, (b) LH and (c) DH stations.**


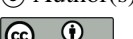



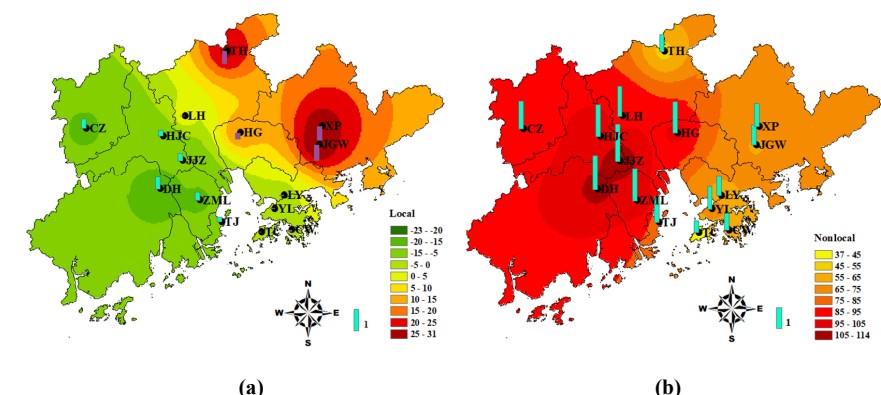

750             **(a)**                       **(b)**

**Fig 8. Spatial distribution of ozone contribution from (a) local and (b) non-local emissions (µg m⁻³) of each station and their annual change rate in the Pearl River Delta. Bars in blue above / in purple below the station point indicate increasing / decreasing contributions. The bar length in the legend corresponds to an annual increase of 1 µg m⁻³. Ozone contributions from local emissions show positive but decreasing trend in**

**the northeastern and negative but increasing trend in the southwestern. Ozone contributions from non-local emissions are positive and increasing region-wide.**

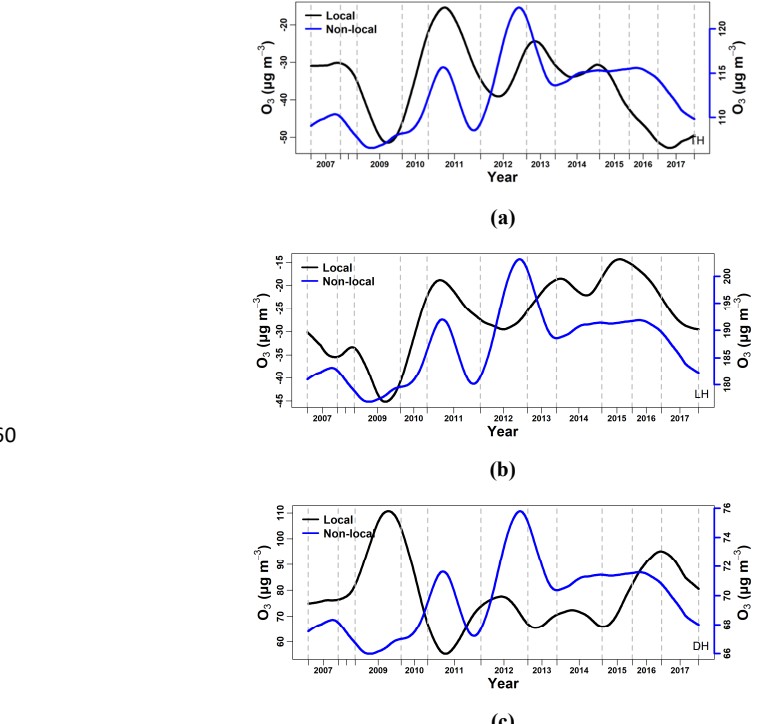


**Fig 9. Long-term trend of ozone contributed by local (black) and non-local (blue) emission sources from**

**2007 to 2017 at (a) TH, (b) LH and (c) DH stations during ozone episodes. Inver**





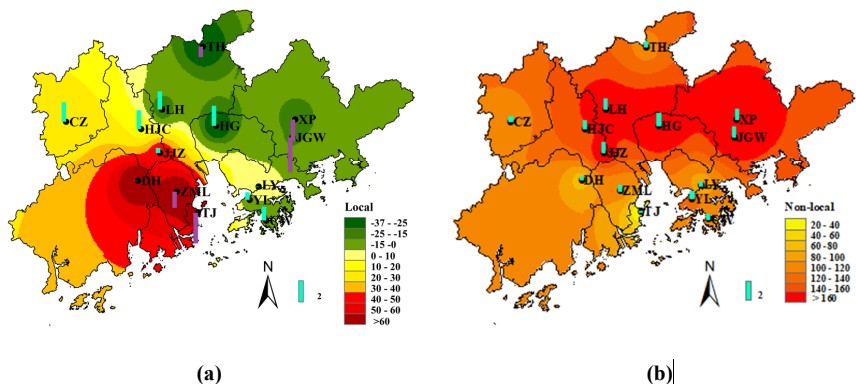

(a)             (b)

**Fig 10. Spatial distribution of ozone contribution from (a) local and (b) non-local emissions (µg m⁻³) of each station and their annual change rate in the Pearl River Delta during ozone episodes. Bars in blue above / in purple below the station point indicate increasing / decreasing contributions trend. The bar length in the legend corresponds to an annual increase of 2 µg m⁻³. Ozone contributions from local emissions are positive in the southwestern and negative in the northeastern. Central PRD is the only area with increasing local ozone contribution trend. Ozone contributions from non-local emissions are positive and increasing region-wide.**



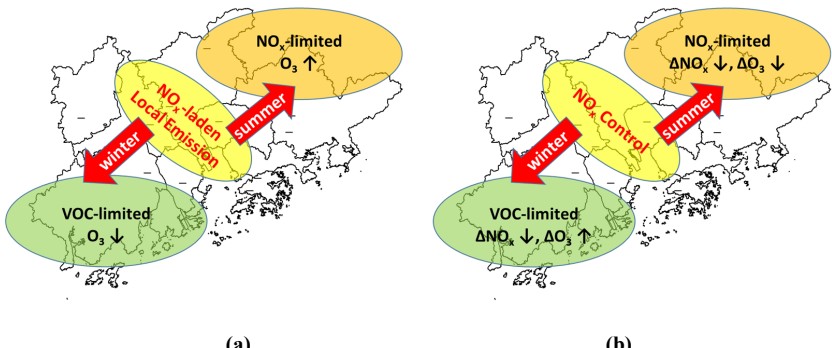

(a)             (b)

**Fig 11. A conceptual diagram on the impacts of local emissions on (a) ozone concentrations and (b) their changing trends over the Pearl River Delta. Local NOₓ-laden emissions increase ozone level (O₃ ↑) in the downwind northeastern in summer and fall, but the increase is suppressed due to the preferential NOₓ control (Δ NOₓ ↓), leading to net ozone decrease (Δ O₃ ↓). In comparison, local emissions decrease ozone level (O₃ ↓) in the downwind southwestern in winter and spring, but the decrease is also mitigated due to NOₓ control (Δ NOₓ ↓), leading to net ozone increase (Δ O₃ ↑). Such a phenomenon is essentially governed by different ozone formation regimes in northeastern (NOₓ-limited) and southwestern PRD (VOC-limited).**




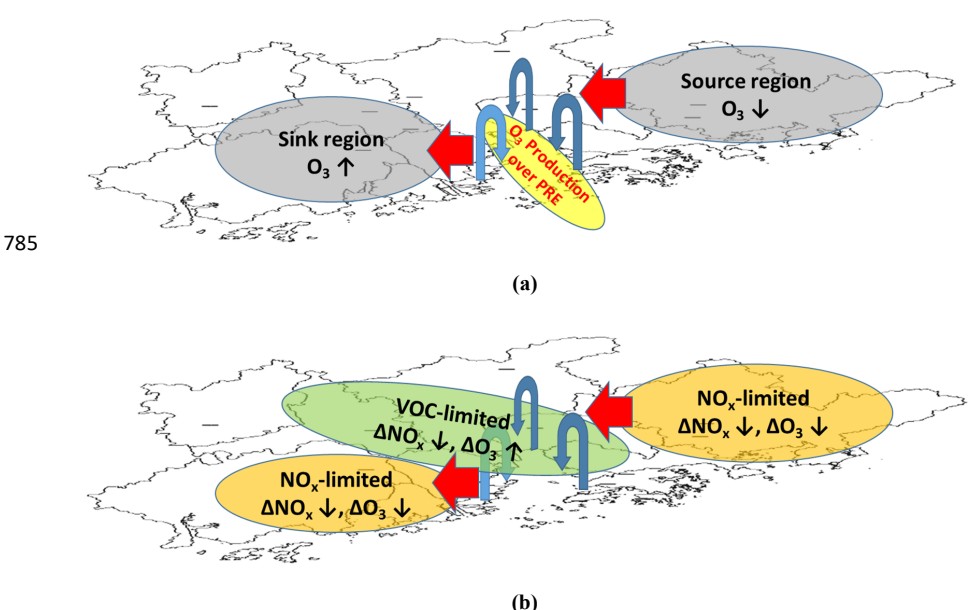


**(a)**

**(b)**

**Fig 12. A conceptual diagram on the impact of local emissions on (a) ozone concentrations and (b) their**
**changing trends in the Pearl River Delta during ozone pollution episodes. The blue curved angles indicate**
**micro-scale circulations such as land-sea breeze developed around the Pearl River Estuary (PRE). The**
**micro-scale circulation leads to a high ozone area around the PRE due to effective mixing and reaction**
**between VOCs and NOx. High ozone is transported to southwestern by easterly wind, increasing local ozone**
**contribution. In comparison, northeastern is a source region of ozone therefore its local contribution is**
**negative. With higher biogenic VOC emissions and VOC oxidation rate during ozone episodes, most of PRD**
**is in the NOx-limited ozone formation regime except for urban central PRD which is still VOC-limited due**
**to intense NOx emissions. Therefore, reduced NOx emissions ( Δ NOx ↓ ) lead to decreasing ozone level ( Δ O₃**
**↓ ) over both northeastern and southwestern and increasing ozone level over central PRD ( Δ O₃ ↑ ).**