# Peer review of "Quantitative impacts of meteorology and precursor emission changes on the longterm trend of ambient ozone over the Pearl River Delta, China and implications for ozone control strategy"

_Atmospheric Chemistry and Physics, 2019_

## Referee Comment (RC1) · Anonymous Referee #1 · 14 May 2019

General comments: Ozone (O3), as a criteria air pollutant, is attracting increasing concerns in China due to the rapid rise in concentrations across the country. This paper studied the O3 pollution in an economically boomed region of China (PRD in southern China), in terms of the meteorological impacts, local contribution and regional transport. The statistical methods were used, with interesting and meaningful results being reported. Basically, the decadal changes of O3 in PRD were well explained, except for some aspects where further clarifications or reorganizations are needed. Though some of the findings were already known knowledge, the paper further consolidate our

understandings and fully demonstrated its value in future O3 pollution control in this region. Thus, the paper is recommended to be accepted after the following problems are addressed. First, I do not quite agree with the authors' statement of the "conceptual model". Generally, a conceptual model is established based on some phenomena, and is further verified by the results. In this study, I would like to suggest the authors to replace the "conceptual model" with the "discussions" on the results, because I did not see the verification of the "model". Second, the discussions on the O3 episodes were relatively weak and the trend analyses did not seem to be appropriate for O3 episodes, due to the limited and inconsistent number of episode days in every years. This section needs to be reorganized and some discussions should be clarified or corrected. Third, the discussions on O3 pollution in 2016 and 2017 look a bit weird, which should be reorganized. Lastly, it will be good if the O3 distribution, trend and the influencing factors can be discussed separately by seasons. In fact, the increases of springtime O3 in PRD in recent years were striking, in contrast to the overall unchanged O3 in summer and autumn. The paper would be more informative with the discussions on O3 pollution in different seasons and a special focus on the season when O3 is highest or increasing with the highest rates.

Specific comments: 1. Page 6, line 233-234. What were the reasons of the minor changes in O3 due to emissions during 2011 - 2015, in contrast to the significant increase before 2011? Throughout the paper, the changes in the meteorological and artificial impacts, especially the turning points of O3 variations, should be discussed. 2. Page 6, lines 236-238. The strong statement must be evidenced. This statement is actually contradictory to the later finding that "meteorological adjustment does not alter ozone concentration much" (page 9, lines 362 – 363) on episode days. Please clarify. 3. Page 6, lines 259 – 260. Please briefly explain why central and western PRD were the regions that were most sensitive to meteorological conditions in O3 pollution? 4. Page 7, line 267. ". . .and that most local emissions are concentrated in the central PRD area". Add references to support the statement. 5. Page 7, lines 303 – 305. The reasons for picking the three sites should be given. It will be better to plot all the sites in

the supplement. 6. Page 8, lines 349 – 351. Change "strengthening" to "constraining" or "restraining". Why only VOCs should be controlled? Also, cutting VOCs emissions will not prevent the decrease of NO titration to O3. The statements should be more accurate throughout. 7. Page 9, lines 377 – 378. What were the causes of levelling-off and decrease of non-local contributions? As commented above, the changes are worth to be discussed, which may relate to the nationwide emission controls. 8. Page 11, lines 448 – 450. It is most likely that O3 formation in the northeastern PRD became more limited by NOx, however evidences should be provided to prove the shift of O3 formation regime from VOC-limited to NOx-limited in the southwestern. 9. Page 11, lines 455 – 464. The discussions on O3 episodes need to be deepened. For example, the winds were not always from the east during O3 episodes, which in fact were from the northeast with low speeds in most cases under continental anticyclones, and from the northwest with the approaching of tropical cyclones. I do not think that the winds during O3 episodes can be simplified as easterly, so did the other characteristics which were discussed as an integration in this paper. 10. Page 13, lines 529 – 530. The term "optimal effective NOx/VOC ratio" needs to be annotated. The optimal ratio in fact means highest O3 production rates, which is the worst from the angle of O3 pollution control.

---

## Author Comment (AC1) · 22 May 2019

Response to the Comments from Referee #1

General comments: Ozone ($O_3$), as a criteria air pollutant, is attracting increasing concerns in China due to the rapid rise in concentrations across the country. This paper studied the $O_3$ pollution in an economically boomed region of China (PRD in southern China), in terms of the meteorological impacts, local contribution and regional transport. The statistical methods were used, with interesting and meaningful results being reported. Basically, the decadal changes of $O_3$ in PRD were well explained, except for some aspects where further clarifications or reorganizations are needed. Though some of the findings were already known knowledge, the paper further consolidate our understandings and fully demonstrated its value in future $O_3$ pollution control in this region. Thus, the paper is recommended to be accepted after the following problems are addressed.

1. First, I do not quite agree with the authors' statement of the "conceptual model". Generally, a conceptual model is established based on some phenomena, and is further verified by the results. In this study, I would like to suggest the authors to replace the "conceptual model" with the "discussions" on the results, because I did not see the verification of the "model".
Response: According to Pun and Seigneur (1999), a conceptual model is a qualitative compilation of the physical and chemical processes that govern the formation of pollutants, which, to the extent possible, is supported by quantitative information. Conceptual model helps to elucidate pollutant contributions from local and non-local sources and their temporal and spatial variations, and the underlying reasons governing such variations. In this study, we use information about ozone precursor (VOCs and $NO_x$) emissions and their changing trends, ozone formation regimes, and the monsoonal and micro-scale synoptic conditions over different sub-regions of the Pearl River Delta to explain the spatiotemporal variations of modeling object (locally formed $O_3$ with meteorological adjustment). Therefore, we believe our analysis approach could be called a 'conceptual model' and would like to keep using it. Good interpretation on the modeling object (locally formed $O_3$ with meteorological adjustment) both under general conditions and during episodes serves as verification of the applicability of the conceptual model.

2. Second, the discussions on the O3 episodes were relatively weak and the trend analyses did not seem to be appropriate for O3 episodes, due to the limited and inconsistent number of episode days in every years. This section needs to be reorganized and some discussions should be clarified or corrected.
Response: See below responses to specific comments. We will clarify and correct some discussions in the revision.

3. Third, the discussions on O3 pollution in 2016 and 2017 look a bit weird, which should be reorganized.
Response: The logic here is that in previous sections we have examined the long-term trends of meteorological impacts, and $O_3$ contributions from non-local emissions and different factors of local emissions. Our statistical analysis framework also has the capability of quantitatively separating the impact of meteorology and local and non-local emissions. Actually we can do this year by year. Considering the length of the paper, we decide to show this capability in the year of 2016 and 2017, the most recent two years in the study period. We will reorganize this part in the

revision to highlight our objective.

4. Lastly, it will be good if the O3 distribution, trend and the influencing factors can be discussed separately by seasons. In fact, the increases of springtime O3 in PRD in recent years were striking, in contrast to the overall unchanged O3 in summer and autumn. The paper would be more informative with the discussions on O3 pollution in different seasons and a special focus on the season when O3 is highest or increasing with the highest rates.

Response: The main point of this paper is to quantitatively examine the impacts of meteorology and precursor emissions from within and outside the PRD on the evolution of ozone during the past decade by using a statistical analysis framework combining meteorological adjustment and source apportionment and a conceptual model. Seasonal variation is indeed a very interesting issue but cannot be addressed in detail here due to the length of the paper. We will investigate it in our further study.

Specific comments:

1. Page 6, line 233-234. What were the reasons of the minor changes in O3 due to emissions during 2011 - 2015, in contrast to the significant increase before 2011? Throughout the paper, the changes in the meteorological and artificial impacts, especially the turning points of O3 variations, should be discussed.

Response: According to the PRD emission inventory developed by our research group (Figure 1 below, manuscript under preparation), increase in VOCs emission started to mitigate in 2011, while NOx emissions showed significant reduction starting from 2013. As PRD is generally in a VOC-limited ozone formation regime, reduction in the magnitude of VOCs emission increase is likely responsible for the minor changes in O$_3$ during 2011-2015. We will add this point in the revision.

[Figure]

Figure 1 Emissions of NOx and VOCs in the Pearl River Delta during 2006-2016

2. Page 6, lines 236-238. The strong statement must be evidenced. This statement is actually contradictory to the later finding that "meteorological adjustment does not alter ozone concentration much" (page 9, lines 362 – 363) on episode days. Please clarify.

Response: The statement in lines 236-238 is based on Fig. 3b which clearly shows that the black ozone spikes are shortened to blue after meteorological adjustment. This indicates that meteorology plays a leading role in generating ozone episodes. It should be noted that meteorological adjustment here is against the average meteorological condition during the entire period. In comparison, the meteorological adjustment in the episodic analysis is against the average meteorological condition during episode days. Although the average meteorological condition varies greatly in different years, that during episodes does not differ much in different

years. Therefore, we conclude that "meteorological adjustment does not alter ozone concentration much" on episode days.

3. Page 6, lines 259 – 260. Please briefly explain why central and western PRD were the regions that were most sensitive to meteorological conditions in O3 pollution?

Response: Generally, area with higher pollutant emission is more sensitive to changes in meteorological condition (Seo et al., 2014). Various studies have shown that central and western PRD is the area with the most intense $O_3$ precursor (VOCs and NOx) emissions over the PRD (e.g. Zheng et al., 2009) therefore is more sensitive to meteorological condition. This information will be added in the revision.

4. Page 7, line 267. "…and that most local emissions are concentrated in the central PRD area". Add references to support the statement.

Response: We will revise this statement as "…and that most local emissions are concentrated in the central and western PRD area". Reference Zheng et al. (2009) will be added in the revision.

5. Page 7, lines 303 – 305. The reasons for picking the three sites should be given. It will be better to plot all the sites in the supplement.

Response: There is a north-south gradient in the spatial distributions of all three principal component loadings, as shown in Fig. 5a. Therefore, we select Tianhu (TH) in the north, Luhu (LH) in central, and Donghu (DH) in the south of PRD to study the long-term trend of ozone contributed by local and non-local emission sources in different areas. Ozone contributions at the other stations will be added in the supplement.

6. Page 8, lines 349 – 351. Change "strengthening" to "constraining" or "restraining". Why only VOCs should be controlled? Also, cutting VOCs emissions will not prevent the decrease of NO titration to O3. The statements should be more accurate throughout.

Response: Here we are talking about "strengthening local VOC emission control". There is a "control" at the end of the sentence, therefore we cannot change "strengthening" to "constraining". In addition, NO titration of $O_3$ is dependent on VOC/$NO_x$ ratio. If the ratio is lower, NO titration of $O_3$ near emission sources would be higher. Therefore, controlling VOCs emission to reduce VOC/$NO_x$ ratio would to some extent enhance NO titration so as to reduce $O_3$ level.

7. Page 9, lines 377 – 378. What were the causes of levelling off and decrease of non-local contributions? As commented above, the changes are worth to be discussed, which may relate to the nationwide emission controls.

Response: At this time we are not aware what the exact reason for levelling off and decrease of non-local contribution after 2014. We look at VOCs and $NO_x$ emission inventory over non-PRD area in Guangdong Province developed by our research group (Figure 2 below, manuscript under preparation). $NO_x$ emissions decreases and VOCs emissions increases after 2014. As most of the non-PRD area in Guangdong may in a transitional ozone formation regime in an annual average, such a slight change in $NO_x$ and VOC emissions lead to little changes in $O_3$ formation and transport into the PRD. We must emphasize that the above argument is simply our speculation and further modeling study is needed to explain such a phenomenon.

[Figure]

Figure 2 Emissions of NOx and VOCs in the non-Pearl River Delta of Guangdong Province during 2006-2015

8. Page 11, lines 448 – 450. It is most likely that O3 formation in the northeastern PRD became more limited by NOx, however evidences should be provided to prove the shift of O3 formation regime from VOC-limited to NOx-limited in the southwestern.

Response: There is no publication reporting the shift in $O_3$ formation regime in southwestern PRD, as such a shift mainly occurs in recent years. However, we may speculate such a shift during pollution episodes considering the intense biogenic VOC emissions over southwestern rural area when temperature is high and solar radiation is strong. Jin and Holloway (2015) discovered that $O_3$ photochemistry is NOx limited from April to October and transitional or mixed in other months using satellite observed $HCHO/NO_2$ ratio. Although there is large uncertainty by using $HCHO/NO_2$ to infer ozone formation regime, such a trend of shifting to $NO_x$-limited regime during ozone episodes is evidenced.

9. Page 11, lines 455 – 464. The discussions on O3 episodes need to be deepened. For example, the winds were not always from the east during O3 episodes, which in fact were from the northeast with low speeds in most cases under continental anticyclones, and from the northwest with the approaching of tropical cyclones. I do not think that the winds during O3 episodes can be simplified as easterly, so did the other characteristics which were discussed as an integration in this paper.

Response: In comparison with general conditions, prevailing winds during episodes are more easterly, as shown in Fig. S5 (also provided below). We are not saying that the winds were always from the east during ozone episodes. Slight shift to northeasterly or southeasterly would not change the conclusion from the conceptual model that southwestern PRD around DH, ZML and TJ stations is a sink region of ozone produced around the Pearl River Estuary, thereby having high ozone levels during episodes. Northwest wind by the approaching of tropical cyclones could bring ozone episodes to HK and Shenzhen as they are located downwind of central PRD. However, from an entire region point of view, northwest wind during ozone episodes is rather limited.

[Figure]

Fig S5. Wind rose under general conditions (left) and during ozone episodes (right) during 2007 and 2017 in the Pearl River Delta.

10. Page 13, lines 529 – 530. The term "optimal effective NOx/VOC ratio" needs to be annotated. The optimal ratio in fact means highest O3 production rates, which is the worst from the angle of O3 pollution control.

Response: Here we are exactly talking about $NO_x$/VOC ratio with highest $O_3$ production rate. Our previous publication (Ou et al., 2016) concluded that in the VOC-limited regime, $O_3$ reduction can be possible in short-term but cannot reduce $O_3$ into attainment. The only way is to control $NO_x$ emission so as to change ozone formation regime from VOC-limited to $NO_x$-limited. In this process, $O_3$ concentration would increase slightly in the beginning, and drop significantly after bypassing the turning point with the optimal effective $NO_x$/VOC ratio that corresponds with the highest $O_3$ level.

References

1.  Pun, B.K. and C. Seigneur: Understanding particulate matter formation in the California San Joaquin Valley: conceptual model and data needs, Atmospheric Environment, 33, 4865-4875, 1999.

2.  Seo, J., Youn, D., Kim, J. Y., and Lee, H.: Extensive spatio-temporal analyses of surface ozone and related meteorological variables in South Korea for 1999-2010, Atmospheric Chemistry and Physics,14,12(2014-06-26), 14, 1191-1238, doi: 10.5194/acpd-14-1191-2014, 2014.

3.  Zheng, J., Shao, M., Che, W. W., Zhang, L., Zhong, L., Zhang, Y., and Streets, D.: Speciated VOC emission inventory and spatial patterns of ozone formation potential in the Pearl River Delta, China, Environmental Science & Technology, 43, 8580-8586, doi: 10.1021/es901688e, 2009.

4.  Ou, J., Yuan, Z., Zheng, J., Huang, Z., Shao, M., Li, Z., Huang, X., Guo, H., and Louie, P. K. K.: Ambient Ozone Control in a Photochemically Active Region: Short-Term Despiking or Long-Term Attainment?, Environmental Science & Technology, 50, 5720, doi: 10.1021/acs.est.6b00345, 2016.

5.  Jin, X. and Holloway, T.: Spatial and temporal variability of ozone sensitivity over China observed

from the Ozone Monitoring Instrument. Journal of Geophysical Research Atmospheres, 2015, 120(14), doi: 10.1002/2015JD023250.

---

## Referee Comment (RC2) · Anonymous Referee #2 · 22 Jun 2019

The manuscript by Yang et al. describes the quantitatively statistical analysis of meteorology and precursor emissions impacts on ozone evolution in PRD region. The strategy of the analysis is clear and the manuscript has been relatively well organized for such a topic. This study presents an innovative approach to quantify the contributions of meteorological factors as well as non-local precursor emissions for ozone concentration in PRD. Besides, this study reveals the special distribution differences of ozone increase contribution under general conditions and ozone episodes, and uses a model to explain the difference result from emission control. Furthermore, specific

suggestions are given in order to acquire high ozone reduction efficiency. Therefore it merits to be published in ACP. However, more detailed explanations are expected to make this manuscript complete and more convincing. In the following, I had a number of specific comments for the authors' reference to address before publication.

Specific comments:

1. It is confusing for the final sentence at the second paragraph of section 3.1. How to conclude the meteorological condition is the most important driving factor for ozone episodes through Fig. 3b?

2. In Fig. 4a, what is the legend of the color contour?

3. In the first paragraph of section 3.3, considering the monsoon climate, why local emissions have distinct impacts while non-local emissions are consistent?

4. In section 3.3, I would like to suggest the authors to further explain how to distinguish the non-local emissions from the impact of meteorological condition, as the PC1 is associated with northeasterly wind and meteorology has been removed in meteorological adjustment part.

5. What does the score mean in Fig. 5b and Fig. 6? I would like to suggest the authors add more explanations in figure caption for Fig. 5b and Fig. 6.

6. As SSR plays an important role in meteorology factors, I wonder if it is necessary to add the correlation of SSR with PCs?

7. In sections 3.3 and 3.5, long-term trends of local and non-local emission contributions on ozone is an interesting finding. The authors are suggested to provide in-depth discussion the influencing factors shaping the trends rather than simply describing them, both under general conditions and during ozone episodes.

8. There exist some uncommon usages of scientific writing English in the manuscript, such as Line 74 'such a philosophy' and Line 90 'is suffered from'.

9. In Line 127, u and v are wind direction and speed respectively while in Fig. 6 u and v both represent wind direction.

10. Some grammatical mistakes should be corrected, for example in Line 299, there should be a sentence after the word 'while'.

---

## Author Comment (AC2) · 3 Jul 2019

**Response to the Comments from Referee #1**

General comments: Ozone (O3), as a criteria air pollutant, is attracting increasing concerns in China due to the rapid rise in concentrations across the country. This paper studied the O3 pollution in an economically boomed region of China (PRD in southern China), in terms of the meteorological impacts, local contribution and regional transport. The statistical methods were used, with interesting and meaningful results being reported. Basically, the decadal changes of O3 in PRD were well explained, except for some aspects where further clarifications or reorganizations are needed. Though some of the findings were already known knowledge, the paper further consolidate our understandings and fully demonstrated its value in future O3 pollution control in this region. Thus, the paper is recommended to be accepted after the following problems are addressed.

1. First, I do not quite agree with the authors' statement of the "conceptual model". Generally, a conceptual model is established based on some phenomena, and is further verified by the results. In this study, I would like to suggest the authors to replace the "conceptual model" with the "discussions" on the results, because I did not see the verification of the "model".

Response: According to Pun and Seigneur (1999), a conceptual model is a qualitative compilation of the physical and chemical processes that govern the formation of pollutants, which, to the extent possible, is supported by quantitative information. Conceptual model helps to elucidate pollutant contributions from local and non-local sources and their temporal and spatial variations, and the underlying reasons governing such variations. In this study, we use information about ozone precursor (VOCs and NOx) emissions and their changing trends, ozone formation regimes, and the monsoonal and micro-scale synoptic conditions over different sub-regions of the Pearl River Delta to explain the spatiotemporal variations of modeling object (locally formed O3 with meteorological adjustment). Therefore, we believe our analysis approach could be called a 'conceptual model' and would like to keep using it. Good interpretation on the modeling object (locally formed O3 with meteorological adjustment) both under general conditions and during episodes serves as verification of the applicability of the conceptual model.

2. Second, the discussions on the O3 episodes were relatively weak and the trend analyses did not seem to be appropriate for O3 episodes, due to the limited and inconsistent number of episode days in every years. This section needs to be reorganized and some discussions should be clarified or corrected.

Response: I don't know why the reviewer thinks the trend analysis did not seem to be appropriate for  $O_3$  episodes. Yes the number of episodic days are different across the 11 years, such a difference can be largely accounted and removed by meteorological adjustment. In this section, we add explanation on the slight changes of non-local emission contribution on ozone after 2014.

**3. Third, the discussions on O3 pollution in 2016 and 2017 look a bit weird, which should be reorganized.**

Response: The logic here is that in previous sections we have examined the long-term trends of meteorological impacts, and  $O_3$  contributions from non-local emissions and different factors of local emissions. Our statistical analysis framework also has the capability of quantitatively separating the impact of meteorology and local and non-local emissions. Actually we can do this

year by year. Considering the length of the paper, we decide to show this capability in the year of 2016 and 2017, the most recent two years in the study period.

4. Lastly, it will be good if the O3 distribution, trend and the influencing factors can be discussed separately by seasons. In fact, the increases of springtime O3 in PRD in recent years were striking, in contrast to the overall unchanged O3 in summer and autumn. The paper would be more informative with the discussions on O3 pollution in different seasons and a special focus on the season when O3 is highest or increasing with the highest rates.

Response: The main point of this paper is to quantitatively examine the impacts of meteorology and precursor emissions from within and outside the PRD on the evolution of ozone during the past decade by using a statistical analysis framework combining meteorological adjustment and source apportionment and a conceptual model. Seasonal variation is indeed a very interesting issue but cannot be addressed in detail here due to the length of the paper. We will investigate it in our further study.

Specific comments:

 Page 6, line 233-234. What were the reasons of the minor changes in O3 due to emissions during 2011 - 2015, in contrast to the significant increase before 2011? Throughout the paper, the changes in the meteorological and artificial impacts, especially the turning points of O3 variations, should be discussed.

Response: According to the PRD emission inventory developed by our research group (Figure S1, also provided as Figure 1 below, manuscript under preparation), increase in VOCs emission started to mitigate in 2011, while NOx emissions showed significant reduction starting from 2013. As PRD is generally in a VOC-limited ozone formation regime, reduction in the magnitude of VOCs emission increase is likely responsible for the minor changes in O3 during 2011-2015. This is added in lines 230-234 in the revised manuscript.

Fig 1 Emissions of NOx and VOCs in the Pearl River Delta during 2006-2016

2. Page 6, lines 236-238. The strong statement must be evidenced. This statement is actually contradictory to the later finding that "meteorological adjustment does not alter ozone concentration much" (page 9, lines 362 – 363) on episode days. Please clarify.

Response: Agree with the reviewer that we cannot draw the conclusion that meteorological condition is "the most" important driving factor. We change it to "one of the most" in lines 243-

244 in the revised manuscript, as it can be clearly evidenced by Fig. 3(b) that ozone spikes were shortened after meteorological adjustment. This indicates that meteorology plays a leading role in generating ozone episodes. It should be noted that meteorological adjustment here is against the average meteorological condition during the entire period. In comparison, the meteorological adjustment in the episodic analysis is against the average meteorological condition during episode days. Although the average meteorological condition varies greatly in different years, that during episodes does not differ much in different years. Therefore, we conclude that "meteorological adjustment does not alter ozone concentration much" on episode days.

3. Page 6, lines 259 – 260. Please briefly explain why central and western PRD were the regions that were most sensitive to meteorological conditions in O3 pollution?

Response: Generally, area with higher pollutant emission is more sensitive to changes in meteorological condition (Seo et al., 2014). Various studies have shown that central and western PRD is the area with the most intense O3 precursor (VOCs and NOx) emissions over the PRD (e.g. Zheng et al., 2009a) therefore is more sensitive to meteorological condition. This is added in lines 266-269 in the revised manuscript.

4. Page 7, line 267. "...and that most local emissions are concentrated in the central PRD area". Add references to support the statement.

Response: We revise this statement as "...and that most local emissions are concentrated in the central and western PRD area". Reference Zheng et al. (2009a) is added in the revision.

5. Page 7, lines 303 - 305. The reasons for picking the three sites should be given. It will be better to plot all the sites in the supplement.

Response: There is a north-south gradient in the spatial distributions of all three principal component loadings, as shown in Fig. 5a. Therefore, we select Tianhu (TH) in the north, Luhu (LH) in central, and Donghu (DH) in the south of PRD to study the long-term trend of ozone contributed by local and non-local emission sources in different areas. Ozone contributions at the other stations is added in Fig. S2 of the supplement.

6. Page 8, lines 349 – 351. Change "strengthening" to "constraining" or "restraining". Why only VOCs should be controlled? Also, cutting VOCs emissions will not prevent the decrease of NO titration to O3. The statements should be more accurate throughout.

Response: Here we are talking about "strengthening local VOC emission control". There is a "control" at the end of the sentence, therefore we cannot change "strengthening" to "constraining". In addition, NO titration of O3 is dependent on VOC/NOx ratio. If the ratio is lower, NO titration of O3 near emission sources would be higher. Therefore, controlling VOCs emission to reduce VOC/NOx ratio would to some extent enhance NO titration so as to reduce O3 level.

7. Page 9, lines 377 - 378. What were the causes of levelling off and decrease of non-local contributions? As commented above, the changes are worth to be discussed, which may relate to the nationwide emission controls.

Response: At this time we are not aware what the exact reason for levelling off and decrease of non-local contribution after 2014. We look at VOCs and NOx emission inventory over non-PRD area in Guangdong Province developed by our research group (Figure 1). NOx emissions decreases and

VOCs emissions increases after 2014. As most of the non-PRD area in Guangdong may in a transitional ozone formation regime in an annual average, such a slight change in NOx and VOC emissions lead to little changes in O3 formation and transport into the PRD. We must emphasize that the above argument is simply our speculation and further modeling study is needed to explain such a phenomenon.

8. Page 11, lines 448 – 450. It is most likely that O3 formation in the northeastern PRD became more limited by NOx, however evidences should be provided to prove the shift of O3 formation regime from VOC-limited to NOx-limited in the southwestern.

Response: There is no publication reporting the shift in O3 formation regime in southwestern PRD, as such a shift mainly occurs in recent years. However, we may speculate such a shift during pollution episodes considering the intense biogenic VOC emissions over southwestern rural area when temperature is high and solar radiation is strong. Jin and Holloway (2015) discovered that O3 photochemistry is NOx limited from April to October and transitional or mixed in other months using satellite observed HCHO/NO2 ratio. Although there is large uncertainty by using HCHO/NO2 to infer ozone formation regime, such a trend of shifting to NOx-limited regime during ozone episodes is evidenced.

9. Page 11, lines 455 – 464. The discussions on O3 episodes need to be deepened. For example, the winds were not always from the east during O3 episodes, which in fact were from the northeast with low speeds in most cases under continental anticyclones, and from the northwest with the approaching of tropical cyclones. I do not think that the winds during O3 episodes can be simplified as easterly, so did the other characteristics which were discussed as an integration in this paper.

Response: In comparison with general conditions, prevailing winds during episodes are more easterly (Figure S6, also provided as Figure 2 below, manuscript under preparation),. We are not saying that the winds were always from the east during ozone episodes. Slight shift to northeasterly or southeasterly would not change the conclusion from the conceptual model that southwestern PRD around DH, ZML and TJ stations is a sink region of ozone produced around the Pearl River Estuary, thereby having high ozone levels during episodes. Northwest wind by the approaching of tropical cyclones could bring ozone episodes to HK and Shenzhen as they are located downwind of central PRD. However, from an entire region point of view, northwest wind during ozone episodes is rather limited.

Fig 2. Wind rose under general conditions (left) and during ozone episodes (right) during 2007 and 2017 in the Pearl River Delta.

10. Page 13, lines 529 – 530. The term "optimal effective NOx/VOC ratio" needs to be annotated. The optimal ratio in fact means highest O3 production rates, which is the worst from the angle of O3 pollution control.

Response: Here we are exactly talking about NOx/VOC ratio with the highest O3 concentration. This is added in lines 496-497 of the revised manuscript. Our previous publication (Ou et al., 2016) concluded that in the VOC-limited regime, O3 reduction can be possible in short-term but cannot reduce O3 into attainment. The only way is to control NOx emission so as to change ozone formation regime from VOC-limited to NOx-limited. In this process, O3 concentration would increase slightly in the beginning, and drop significantly after bypassing the turning point with the optimal effective NOx/VOC ratio that corresponds with the highest O3 level.

[revised manuscript text omitted]

---

## Author Response (AR2)

Dear Prof. Ronald Cohen,

We would like to submit a revision to the manuscript entitled "**Quantitative impacts of meteorology and precursor emission changes on the long-term trend of ambient ozone over the Pearl River Delta, China and implications for ozone control strategy**" (ACP-2019-355.R2) to *Atmospheric Chemistry and Physics*.

We thank both referees for their positive feedbacks on our manuscript. We did try our best to understand some comments, and provide here a detailed response to all of them. We hope that you would find our response compelling and our revised manuscript meets the quality standard of *Atmospheric Chemistry and Physics*. We appreciate your consideration of our manuscript, and look forward to receiving further comments from the referees. If you have any queries, please contact me via zibing@scut.edu.cn.

Yours sincerely,

Zibing Yuan

Professor, South China University of Technology

**Response to the Comments from Referee #1**

General comments: Ozone ($O_3$), as a criteria air pollutant, is attracting increasing concerns in China due to the rapid rise in concentrations across the country. This paper studied the $O_3$ pollution in an economically boomed region of China (PRD in southern China), in terms of the meteorological impacts, local contribution and regional transport. The statistical methods were used, with interesting and meaningful results being reported. Basically, the decadal changes of $O_3$ in PRD were well explained, except for some aspects where further clarifications or reorganizations are needed. Though some of the findings were already known knowledge, the paper further consolidate our understandings and fully demonstrated its value in future $O_3$ pollution control in this region. Thus, the paper is recommended to be accepted after the following problems are addressed.

1.   First, I do not quite agree with the authors' statement of the "conceptual model". Generally, a conceptual model is established based on some phenomena, and is further verified by the results. In this study, I would like to suggest the authors to replace the "conceptual model" with the "discussions" on the results, because I did not see the verification of the "model".

Response: According to Pun and Seigneur (1999), a conceptual model is a qualitative compilation of the physical and chemical processes that govern the formation of pollutants, which, to the extent possible, is supported by quantitative information. Conceptual model helps to elucidate pollutant contributions from local and non-local sources and their temporal and spatial variations, and the underlying reasons governing such variations. In this study, we use information about ozone precursor (VOCs and $NO_x$) emissions and their changing trends, ozone formation regimes, and the monsoonal and micro-scale synoptic conditions over different sub-regions of the Pearl River Delta to explain the spatiotemporal variations of modeling object (locally formed $O_3$ with meteorological adjustment). Therefore, we believe our analysis approach could be called a 'conceptual model'. Good interpretation on the modeling object (locally formed $O_3$ with meteorological adjustment) both under general conditions and during episodes serves as verification of the applicability of the conceptual model.

**However, to avoid further dispute, we also accept the term 'conceptual diagram' and use it in the revised manuscript. This could be potentially more accurate as major information contributing to our understanding can be well illustrated in a diagram, as shown in Figs. 11 and 12.**

2.   Second, the discussions on the O3 episodes were relatively weak and the trend analyses did not seem to be appropriate for O3 episodes, due to the limited and inconsistent number of episode days in every years. This section needs to be reorganized and some discussions should be clarified or corrected.

Response: **We agree with the reviewer that the trend analysis might be associated with larger uncertainty than the general conditions due to much smaller number of cases. This point is highlighted in lines 387-390 of the revised manuscript.** However, we do believe the episode analysis is indispensable, as the annual Air Quality Objective for ozone in China is defined as the 90th percentile (high end) of maximum daily 8 h average in a particular year. Trends during ozone episodes are thus more linked with ozone pollution control and management and useful for policy-makers. We consider our results appropriate, as it can be well explained by the conceptual diagram in section 3.6.

Apart from adding alerts of larger uncertainties in lines 387-390, we didn't reorganize this section as the logic here is quite clear. In section 3.5, we present ozone levels during episodes first, followed by their long-term trends with/without meteorological adjustment. Afterwards, with local/non-local separation, we discuss the trends of local and non-local contribution separately. Findings here are well connected with the development of conceptual diagram in section 3.6.

3. Third, the discussions on O3 pollution in 2016 and 2017 look a bit weird, which should be reorganized.

Response: We try hard to understand why the reviewer feels that the discussions here are weird, and feel it may be owing to the selection of 2016 and 2017 as representative years for analysis. Actually our statistical analysis framework has the capability of quantitatively identify relative contributions of meteorology and local and non-local emissions to ozone contribution in all years at all stations, but we have no place discuss them in detail. Therefore, **selecting only two years is mostly due to the limitation of space of the manuscript. We make our point clearer in line 363-365 of the revised manuscript.** 2016 and 2017 were selected because they are recent years and have shown significant increase in ozone levels. Policy-makers here in the PRD are very interested in the underlying reasons for such a significant increase, under the backdrop of what they believe effective pollution control within the PRD. Our results affirmed their efforts that such a significant increase during 2016 and 2017 is mainly due to changes in meteorology and non-local emissions.

4. Lastly, it will be good if the O3 distribution, trend and the influencing factors can be discussed separately by seasons. In fact, the increases of springtime O3 in PRD in recent years were striking, in contrast to the overall unchanged O3 in summer and autumn. The paper would be more informative with the discussions on O3 pollution in different seasons and a special focus on the season when O3 is highest or increasing with the highest rates.

Response: The main point of this paper is to quantitatively examine the impacts of meteorology and precursor emissions from within and outside the PRD on the evolution of ozone during the past decade by using a statistical analysis framework combining meteorological adjustment and source apportionment and a discussion diagram. Seasonal variations are indeed a very interesting issue but we didn't address in detail here due to the length of the paper. We will investigate it in our further study.

Specific comments:

1. Page 6, line 233-234. What were the reasons of the minor changes in O3 due to emissions during 2011 - 2015, in contrast to the significant increase before 2011? Throughout the paper, the changes in the meteorological and artificial impacts, especially the turning points of O3 variations, should be discussed.

Response: According to the PRD emission inventory developed by our research group (Figure S1, also provided as Figure 1 below, manuscript under preparation), increase in VOCs emission started to mitigate in 2011, while NOx emissions showed significant reduction starting from 2013. As PRD is generally in a VOC-limited ozone formation regime, reduction in the magnitude of VOCs emission increase is likely responsible for the minor changes in $O_3$ during 2011-2015. These were added in lines 239-244 in the revised manuscript.

[Figure]

Fig 1. Emissions of NOx and VOCs in the Pearl River Delta during 2006-2016

2. Page 6, lines 236-238. The strong statement must be evidenced. This statement is actually contradictory to the later finding that "meteorological adjustment does not alter ozone concentration much" (page 9, lines 362 – 363) on episode days. Please clarify.

Response: Agree with the reviewer that we cannot draw the conclusion that meteorological condition is "the most" important driving factor. We changed it to "one of the most" in line 248 in the revised manuscript, as it can be clearly evidenced by Fig. 3(b) that ozone spikes were shortened after meteorological adjustment. This indicates that meteorology plays an important role in generating ozone episodes. It should be noted that meteorological adjustment here is against the average meteorological condition during the entire period. In comparison, the meteorological adjustment in the episodic analysis is against the average meteorological condition during episode days. Although the average meteorological condition varies greatly in different years, that during episodes does not differ much in different years. Therefore, we conclude that "meteorological adjustment does not alter ozone concentration much" on episode days.

3.    Page 6, lines 259 – 260. Please briefly explain why central and western PRD were the regions that were most sensitive to meteorological conditions in O3 pollution?

Response: Generally, area with higher pollutant emission is more sensitive to changes in meteorological condition (Seo et al., 2014). Various studies have shown that central and western PRD is the area with the most intense $O_3$ precursor (VOCs and NOx) emissions over the PRD (e.g. Zheng et al., 2009a) therefore is more sensitive to meteorological condition. This was added in lines 272-275 in the revised manuscript.

4.    Page 7, line 267. "…and that most local emissions are concentrated in the central PRD area". Add references to support the statement.

Response: We revise this statement as "…and that most local emissions are concentrated in the central and western PRD area", as shown in lines 281-283 of the revised manuscript. Reference Zheng et al. (2009a) is added in the revision.

5.    Page 7, lines 303 – 305. The reasons for picking the three sites should be given. It will be better to plot all the sites in the supplement.

Response: There is a north-south gradient in the spatial distributions of all three principal component loadings, as shown in Fig. 5a. Therefore, we select Tianhu (TH) in the north, Luhu (LH) in central, and Donghu (DH) in the south of PRD to study the long-term trend of ozone contributed by local and non-local emission sources in different areas. Justification on site selection is provided in lines 325-333 of the revised manuscript. Ozone contributions at the other stations were added in Fig. S2 of the supplement.

6.    Page 8, lines 349 – 351. Change "strengthening" to "constraining" or "restraining". Why only VOCs should be controlled? Also, cutting VOCs emissions will not prevent the decrease of NO titration to O3. The statements should be more accurate throughout.

Response: Here we are talking about "strengthening local VOC emission control". There is a "control" at the end of the sentence, therefore we cannot change "strengthening" to "constraining". In addition, NO titration of $O_3$ is dependent on VOC/$NO_x$ ratio. If the ratio is lower, NO titration of $O_3$ near emission sources would be higher. Therefore, controlling VOCs emission to reduce VOC/$NO_x$ ratio would to some extent enhance NO titration so as to reduce $O_3$ level.

7.    Page 9, lines 377 – 378. What were the causes of levelling off and decrease of non-local contributions? As commented above, the changes are worth to be discussed, which may relate to the nationwide emission controls.

Response: At this time we are not aware what the exact reason for levelling off and decrease of non-local contribution after 2014. We look at VOCs and $NO_x$ emission inventory over non-PRD area in Guangdong Province developed by our research group (Figure 1). $NO_x$ emissions decreases and VOCs emissions increases after 2014. As most of the non-PRD area in Guangdong may in a transitional ozone formation regime in an annual average, such a slight change in $NO_x$ and VOC emissions lead to little changes in $O_3$ formation and transport into the PRD. We must emphasize that the above argument is simply our speculation and further modeling study is needed to explain such a phenomenon. This point is added in lines 413-415 of the revised manuscript.

8.    Page 11, lines 448 – 450. It is most likely that O3 formation in the northeastern PRD became more limited by NOx, however evidences should be provided to prove the shift of O3 formation regime from VOC-limited to NOx-limited in the southwestern.

Response: There is no publication reporting the shift in $O_3$ formation regime in southwestern PRD, as such a shift mainly occurs in recent years. However, we may speculate such a shift during pollution episodes considering the intense biogenic VOC emissions over southwestern rural area when temperature is high and solar radiation is strong. Jin and Holloway (2015) discovered that $O_3$ photochemistry is NOx limited from April to October and transitional or mixed in other months using satellite observed $HCHO/NO_2$ ratio. Although there is large uncertainty by using $HCHO/NO_2$ to infer ozone formation regime, such a trend of shifting to $NO_x$-limited regime during ozone episodes is evidenced. This is added in lines 504-508 of the revised manuscript.

9.    Page 11, lines 455 – 464. The discussions on O3 episodes need to be deepened. For example, the winds were not always from the east during O3 episodes, which in fact were from the northeast with low speeds in most cases under continental anticyclones, and from the northwest with the approaching of tropical cyclones. I do not think that the winds during O3 episodes can be simplified as easterly, so did the other characteristics which were discussed as an integration in this paper.

Response: In comparison with general conditions, prevailing winds during episodes are more easterly (Figure S6, also provided as Figure 2 below, manuscript under preparation). We are not saying that the winds were always from the east during ozone episodes. Slight shift to northeasterly or southeasterly would not change the conclusion from the conceptual model that southwestern PRD around DH, ZML and TJ stations is a sink region of ozone produced around the Pearl River Estuary, thereby having high ozone levels during episodes. Northwest wind by the approaching of tropical cyclones could bring ozone episodes to HK and Shenzhen as they are located downwind of central PRD. However, from an entire region point of view, northwest wind during ozone episodes is rather limited.

[Figure]

Fig 2. Wind rose under general conditions (left) and during ozone episodes (right) during 2007 and 2017 in the Pearl River Delta.

10.    Page 13, lines 529 – 530. The term "optimal effective NOx/VOC ratio" needs to be annotated. The optimal ratio in fact means highest O3 production rates, which is the worst from the angle of O3 pollution control.

Response: Here we are exactly talking about $NO_x$/VOC ratio leading to the highest $O_3$ concentration. This was modified in lines 520-521 and 570 of the revised manuscript. Our previous publication (Ou et al., 2016) concluded that in the VOC-limited regime, $O_3$ reduction can be possible in short-term but cannot reduce $O_3$ into attainment. The only way is to control $NO_x$ emission so as to change ozone formation regime from VOC-limited to $NO_x$-limited. In this process, $O_3$ concentration would increase slightly in the beginning, and drop significantly after bypassing the turning point with the optimal effective $NO_x$/VOC ratio that corresponds with the highest $O_3$ level.

Response: We changed "while" to "and" in line 321 of the revised manuscript.

[revised manuscript text omitted]